# Shared-AE: Automatic Identification of Shared Subspaces in High-dimensional Neural and Behavioral Activity

**Daiyao Yi**
Department of Electrical & Computer Engineering
Yale University
New Haven, CT 06520, USA
`daiyao.yi@yale.edu`

**Hao Dong, Michael James Higley**
Department of Neuroscience
Yale University
New Haven, CT 06520, USA
`{hao.dong, m.higley}@yale.edu`

**Anne Churchland**
Department of Neurobiology
University of California, Los Angeles
Los Angeles, CA 90095, USA
`AChurchland@mednet.ucla.edu`

**Shreya Saxena**
Department of Biomedical Engineering
Yale University
New Haven, CT 06520, USA
`shreya.saxena@yale.edu`

## Abstract

Understanding the relationship between behavior and neural activity is crucial for understanding brain function. An effective method is to learn embeddings for interconnected modalities. For simple behavioral tasks, neural features can be learned based on labels. However, complex behaviors, such as social interactions, require the joint extraction of behavioral and neural characteristics. In this paper, we present an autoencoder (AE) framework, called Shared-AE, which includes a novel regularization term that automatically identifies features shared between neural activity and behavior, while simultaneously capturing the unique private features specific to each modality. We apply Shared-AE to large-scale neural activity recorded across the entire dorsal cortex of the mouse, during two very different behaviors: (i) head-fixed mice performing a self-initiated decision-making task, and (ii) freely-moving social behavior amongst two mice. Our model successfully captures both 'shared features', shared across neural and behavioral activity, and 'private features', unique to each modality, significantly enhancing our understanding of the alignment between neural activity and complex behaviors. The original code for the entire Shared-AE framework on Pytorch has been made publicly available at: `https://github.com/saxenalab-neuro/Shared-AE`.

## 1 Introduction

Recent advances in hardware and storage capabilities enable us to obtain comprehensive behavioral recordings of the subject along with the corresponding neural activity from large parts of the brain. It is now widely recognized that understanding the relationship between complex neural activity and high-dimensional behavior is a crucial step in brain research that has historically been underestimated (Pereira et al. (2020); Whiteway et al. (2021)). Understanding this relationship provides insight into how the brain processes information during different behaviors and tasks. An effective approach to achieve this is to learn embeddings for these interconnected modalities, which allows for the identification of patterns within complex datasets.

Current research on learning neural embeddings focuses on simple tasks and is largely based on behavioral labels (Pandarinath et al. (2017); Schneider et al. (2023); Zhou & Wei (2020)). However, when it comes to more complex task-related behaviors and social interactions which cannot be captured by simple labels, understanding relevant features becomes significantly more challenging. Moreover, effectively aligning the features of the behavioral and neural modalities requires careful consideration. Integrating these features requires preserving the unique characteristics of each modality while extracting the shared aspects.

In this article, we propose an Autoencoder-based (AE-based) framework termed Shared-AE, incorporating a novel regularization term designed to identify features common to behavior and neural activity. Autoencoders are particularly well-suited for such tasks, as they capture the underlying structure of the data while ensuring that latent variables retain meaningful features from the input. In our approach, we utilize the Cauchy-Schwarz (CS) divergence to enhance the model's capability to extract shared information across modalities. Additionally, we apply the inverse CS divergence to enable private latent variables to capture features unique to each modality.

Our model effectively captures both 'shared features', which are common across modalities, and 'private features', which are unique to each modality. By clearly partitioning these types of features, our approach provides a comprehensive understanding of which aspects of neural activity align with behavior. Compared with other models such as Shi et al. (2019), Gondur et al. (2024), Sani et al. (2021), and Singh Alvarado et al. (2021), our framework successfully addresses the issue of modality leakage - where features from one modality undesirably influence the latent representations of another. This ensures that the shared features truly reflect across-modality insights, while the private features maintain the distinct characteristics of each data source.

We apply the Shared-AE framework towards understanding two very different datasets: (i) a head-fixed mouse performing a self-initiated decision-making task, and (ii) a freely-moving mouse engaged in social behavior. In both cases, neural activity is recorded using widefield calcium imaging (WFCI), capturing many regions across the dorsal cortex. We find that the model **successfully captures shared subspaces across individual and social behavior**, identifying brain regions that are most aligned with the recorded behavior. Additionally, we **identify aligned motifs between the neural and behavioral modalities using a Hidden Markov Model (HMM)**. Furthermore, Shared-AE allows us to **compare neural alignment when using raw video data as compared to pose estimation and related extracted features**, revealing that the activity of certain brain regions aligns more closely with raw behavioral videos. Lastly, Shared-AE can **generalize beyond two simultaneously recorded modalities** to examine the common subspace across multiple modalities. With the increase in naturalistic behavioral recordings and large-scale neural data, Shared-AE offers an automated and interpretable framework to identify behaviorally-relevant neural activity and neurally-relevant behavioral patterns.

We highlight the novelty of Shared-AE through the following key contributions: **(i) Enhanced Interpretability via Latent Subspace Separation:** Shared-AE introduces a novel approach that explicitly separates shared and private latent spaces, improving interpretability by reducing information leakage between modalities. This design allows robust inference even when data from one modality is unavailable or corrupted during testing. **(ii) Improved Performance on Paired and Unpaired Tasks:** Shared-AE maintains the integrity of latent representations in data from different modalities, even when data from one of the modalities is shuffled during evaluation ('unpaired' task). This robustness is demonstrated by its superior performance on the 2AFC dataset, where Shared-AE outperforms existing models such as Gondur et al. (2024); Sani et al. (2021; 2024); Shi et al. (2019). **(iii) Flexibility with Multiple Modalities and Image Data:** Unlike previous methods, Shared-AE is able to handle complex data types such as raw image data, and is also able to extend to more than three modalities. This capability significantly expands its applicability compared to previously published methods, enabling richer behavioral representations than pose estimation alone. **(iv) Minimizing Distribution Distance Instead of Fitting to Predefined Priors:** Rather than conforming to predefined priors, as done in models such as Yi et al. (2022), Shared-AE minimizes the distances between distributions learned from data. This approach yields more flexible and meaningful latent representations. **(v) Utility for Downstream Tasks and Improved Variance Explained:** The separation of shared and private latent variables ensures that the learned representations are robust and suitable for diverse downstream tasks, while enabling insights into brain-behavior relationships. A detailed discussion on the technical and scientific novelty of Shared-AE can be found in Appendix A.3

## 2 RELATED WORKS

### 2.1 MULTI-MODAL INTEGRATION IN NEUROSCIENCE

Multimodal integration is a rapidly growing area of research within artificial intelligence (AI) and machine learning Baltrušaitis et al. (2017); Steyaert et al. (2023); Brenner et al. (2024); Radford et al. (2021); Shi et al. (2021); Tian et al. (2020); Schneider et al. (2023); Zhang et al. (2020); Lake & Higley (2022); Cardin et al. (2020); Lake et al. (2020); Singh Alvarado et al. (2021); Liu et al.

(2021); Shi et al. (2019); Gondur et al. (2024); Sani et al. (2024). This field aims to combine and analyze data from multiple sources to improve the understanding and performance of AI systems. In neuroscience, multimodal data often refers to different types of recordings, such as fMRI and PET (Zhang et al. (2020); Steyaert et al. (2023)), which provide complementary information about brain activity and function. Additionally, multimodality can encompass both behavioral data and corresponding neural activity, providing a more comprehensive view of brain function.

Recent research, such as Sani et al. (2021; 2024), focuses on using dynamical models to generate behaviorally-relevant and behaviorally-irrelevant neural latent variables. While introducing private latent spaces enhances interpretability, these methods struggle when faced with spurious temporal correlations and high-dimensional data. In contrast, our approach extracts modality-specific shared subspaces for across-modality relationships while maintaining interpretability through an autoencoder framework that reconstructs each modality, enabling scalability to complex behaviors. In Singh Alvarado et al. (2021), the authors introduced a multi-encoder model that fused different modalities using a Multimodal Variational Autoencoder (M-VAE) with a product-of-experts (PoE) approach for modality fusion. Similarly, in the Multimodal Mixture-of-Experts VAE (MM-VAE) (Shi et al. (2019)), the authors employed a mixture-of-experts (MoE) strategy instead of PoE to fuse modalities. However, both these methods result in information leakage between modalities, leading to ambiguities as to the origin of the data in the latent space, drastically reducing the interpretability of the latent space. In Gondur et al. (2024), the authors introduced a Gaussian Process (GP) framework to handle temporal relationships and designed separate latent spaces to capture private features for each modality. However, their approach falls short when dealing with unpaired tasks, as it requires both modalities to be present and aligned during inference. This limitation reduces the model's flexibility and robustness in scenarios where data from one modality may be unavailable, shuffled, or otherwise corrupted.

## 2.2 Learning Embeddings in Neuroscience

Recent advancements in hardware and storage capabilities have significantly enhanced the quality and capacity of behavioral and neural recordings. Consequently, a substantial body of research has focused on extracting lower-dimensional features from these high-dimensional datasets. Learning embeddings involves extracting lower-dimensional features for both behavior and neural activity. In the field of learning behavioral embeddings, pose estimation tools such as Lauer et al. (2021); Pereira et al. (2022) have been broadly applied to track keypoint positions from the behavioral videos, and methods like Luxem et al. (2022); Wiltschko et al. (2015); Berman et al. (2014) utilize these keypoint positions to generate lower-dimensional behavioral features. Other works generate behavioral features directly from videos, for example, Batty et al. (2019) applied VAEs for capturing the animal's behavioral features. Furthermore, Whiteway et al. (2021) and Yi et al. (2022); Klys et al. (2018) produce interpretable latent spaces by constraining the latent distribution in various ways. Due to the high-dimensional nature of neural activity, learning lower-dimensional neural representations is also crucial for uncovering neural dynamics. Models such as Churchland et al. (2012); Sani et al. (2021) apply linear methods to learn interpretable embeddings from neural activity. Nonlinear models like Pandarinath et al. (2017); Zhou & Wei (2020) adopt VAE-based approaches to project behavior onto neural activity. Furthermore, Schneider et al. (2023) uses an encoder-based model and constrains the latent space during behavioral tasks using contrastive learning. However, none of these models have effectively addressed more complex behavioral tasks.

## 3 Methods

### 3.1 Overview

Our goal is to develop representations that capture information shared between different modalities, such as behavior and neural activity. We employ an AE-based model with dual encoders to extract features from each modality independently (Fig. 1). The latent space of each modality is subsequently divided into two subspaces: shared and private. To promote commonality across the modalities for shared latent variables, we regularize these subspaces using the Cauchy-Schwarz (CS) divergence that encourages alignment between different subspaces. Moreover, the private latent variables are constrained using the inverse CS divergence regularization that encourages distinctive subspaces. Finally, the corresponding private latent variables are combined with the shared latent variables to reconstruct the original data in each modality using modality-specific decoders (Fig. 1).

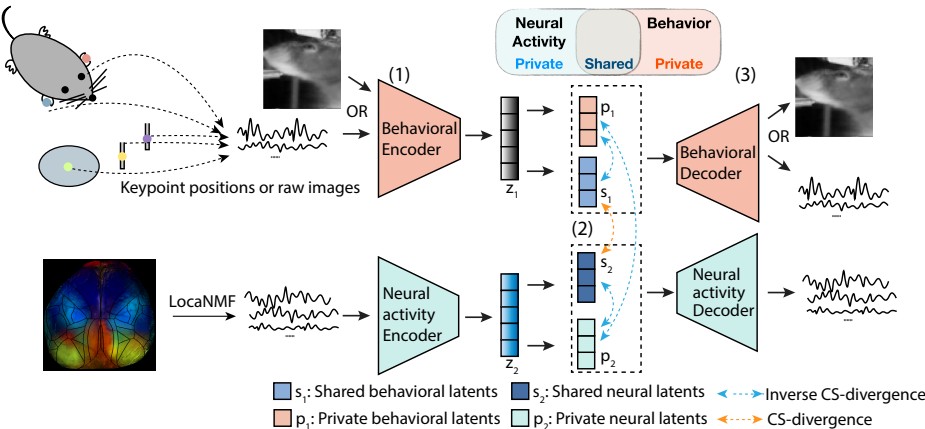

Figure 1: **Shared-AE architecture**: (1) Encoder: each modality is encoded separately. (2) Separation into private vs. shared: after encoding, the latents are separated into private and pre-shared latent variables through linear dense layers. Cauchy-Schwarz (CS) divergence is applied to encourage alignment and the inverse form is applied to encourage orthogonality. (3) Decoder: the latents for each modality are decoded by separate decoders for reconstruction.

## 3.2 MODEL STRUCTURE

We assume that we can record from $C$ different modalities: $i \in \{1, 2, ..., C\}$. Let modality $i$ consist of recorded data $X^i$ over $T$ time points $X^i = \{x_1^i, x_2^i, ..x_T^i\}$. We use a sliding window approach based on the recorded data to capture temporal information in each sample of the input; we define $Y^i = \{y_1^i, y_2^i, ..., y_T^i\}$ as the network input for modality $i$, where $y_t^i = [x_{t-w}^i, x_{t-w+1}^i, ..., x_t^i)]$, with $w$ being the window size. To extract useful representations from each input modality, we apply separate encoders to each modality $f_{\theta^i}$ with a set of learnable parameters $\theta^i$.

Let $z_t^i = f_{\theta^i}(y_t^i)$ represent the encoded representation of the $t$-th sample for modality $i$. $z_t^i$ maps onto two subspaces: the shared latent subspace $s^i$, where $s_t^i = W_s z_t^i + b_s$, and the private latent subspace $p^i$, where $p_t^i = W_p z_t^i + b_p$. Here, $W_s$ and $W_p$ are weights while $b_s$ and $b_p$ are bias terms. Finally, the shared latents and private latents for each modality are concatenated to form a combined latent space $[s^i, p^i]$, which is then decoded back to reconstruct the original input $\hat{y}_t^i = f_{\lambda^i}([s_t^i, p_t^i])$ using separate decoders for each modality. Here, $\hat{y}_t^i$ is the reconstruction and $f_{\lambda^i}$ is the decoder with parameters $\lambda^i$.

In this study, we consider the two (or more) modalities to consist of simultaneously recorded neural activity and behavior. We assume that the behavior is directly recorded using a behavioral video camera, with either the raw video being considered the behavioral modality, or poses that are captured by pose estimation methods such as Lauer et al. (2021); Pereira et al. (2022).

## 3.3 REGULARIZATION ENCOURAGING SHARED VERSUS PRIVATE LATENT VARIABLES

**Two modalities:** To encourage shared structure in the 'shared' latent variables from different modalities, we regularize these using the CS-divergence between $s_t^1$ and $s_t^2$ (Santana et al. (2016); Kampa et al. (2011)). Moreover, to encourage distinct representations in the shared versus private latent variables for each modality, we regularize these using the inverse CS-divergence between the shared latents and the private latents for distinctiveness.

For two probability distribution functions (PDFs) $f_1(x)$ and $f_2(x)$, given the CS inequality (see Appendix A.4), CS-divergence measures the distance between the two distributions (Jenssen et al. (2006)) and is given by:

$$D_{CS}(f_1, f_2) = -\log \frac{\int f_1(x) f_2(x) dx}{\sqrt{\int f_1(x) dx \int f_2(x) dx}} \tag{1}$$

$D_{CS}(f_1, f_2)$ equals zero if and only if the two distributions $f_1(x)$ and $f_2(x)$ are the same. For two given latent subspace with dimension $d$, the $1^{st}$ and $2^{nd}$ modalities, $s_t^1$ and $s_t^2$, $t = 1, .., N$, the PDF of each distribution can be approximated by its Parzen window estimator:

$$\hat{f}_i(s) = \frac{1}{N} \sum_{t=1}^{N} G_{\sigma^2}(s, s_t^i) \tag{2}$$

where $i = 1, 2$; $G_{\sigma^2}$ is the Gaussian kernel with kernel size $\sigma$ (Santana et al. (2016)) and is expressed as $G_{\sigma^2}(s, s_t^i) = \frac{1}{(2\pi\sigma^2)^{\frac{d}{2}}} \exp\{-\frac{||s-s_t^i||^2}{2\sigma^2}\}$. By replacing the actual densities in Eq. 1, the numerator can be rewritten as:

$$\int f_1(s)f_2(s)ds = \int \hat{f}_1(s)\hat{f}_2(s)ds = \frac{1}{N_1 N_2} \sum_{j,k=1}^{N_1,N_2} \int G_{\sigma^2}(s, s_j^1)G_{\sigma^2}(s, s_k^2)ds \tag{3}$$

According to the convolution theorem for Gaussian functions, the above equation can be simplified as $\int f_1(s)f_2(s)ds = \frac{1}{N_1 N_2} \sum_{j,k=1}^{N_1,N_2} G_{(\sqrt{2}\sigma)^2}(s_j^1, s_k^2)$. Here we denote $\sum_{j,k=1}^{N_1,N_2} G_{(\sqrt{2}\sigma)^2}(s_j^1, s_k^2)$ as $V(f_1, f_2)$. Similarly, by replacing the $f_2(s)$ by $f_1(s)$, the expression becomes $\int f_1^2(s)ds = \frac{1}{N_1^2} \sum_{j,j'=1}^{N_1,N_1} G_{(\sqrt{2}\sigma)^2}(s_j^1, s_{j'}^2)$, then the denominator can be written as $\sqrt{(V(f_1, f_1), V(f_2, f_2))}$, for simplification, we put $V(f_i, f_i)$ as $V(f_i)$ where $i \in \{1, 2\}$.

Finally, equation (1) can be expressed as:

$$\mathcal{L}_{CS} := D_{CS}(f_1, f_2) = -\log \frac{V(f_1, f_2)}{\sqrt{V(f_1)V(f_2)}} \tag{4}$$

Here, $f_1(s^1)$ and $f_2(s^2)$ represents the distribution of shared latent space for modality 1 and 2. In Equation (4), minimizing $V(f_1)$ would result in the spreading out of $f_1(x)$, while maximizing $V(f_1, f_2)$ would make the samples in both distributions closer together (Yi et al. (2022)). Thus, we minimize $\mathcal{L}_{CS}$ for shared latent spaces and maximize it to reduce the similarity between shared and private latent spaces within the same modality. Additionally, by maximizing this value, we promote distinctiveness between the private latent spaces of different modalities. There is a close relationship between the VAE and the CS-regularized AE, as detailed in Appendix A.6.

### 3.4 OBJECTIVE FUNCTION FOR TWO MODALITIES
Overall, the objective function can be expressed as:

$$\mathcal{L} = \mathcal{L}_{MSE_1} + \mathcal{L}_{MSE_2} + \alpha\mathcal{L}_{cs_{s_1 s_2}} + \beta\mathcal{L}_{cs_{s_1 p_1}}^{-1} + \gamma\mathcal{L}_{cs_{s_2 p_2}}^{-1} + \delta\mathcal{L}_{cs_{p_1 p_2}}^{-1} \tag{5}$$

Here, the terms $\mathcal{L}_{MSE_1}$ and $\mathcal{L}_{MSE_2}$ represent the reconstruction loss of the two input modalities, respectively. The $\mathcal{L}_{cs_{s_1 s_2}}$, $\mathcal{L}_{cs_{s_1 p_1}}$, $\mathcal{L}_{cs_{s_2 p_2}}$, and $\mathcal{L}_{cs_{p_1 p_2}}$ represents the CS-divergence loss between different latent subspaces. $\alpha$ is introduced to control the similarity between the shared latent space. $\beta, \gamma$, and $\delta$ are adopted to assist the model in producing independent latent subspaces.

### 3.5 GENERALIZATION TO MORE THAN TWO MODALITIES

The CS-divergence can also be extended to measure the distance between multiple distributions. For $C$ number of PDFs, the CS-divergence can be written as follows:

$$\mathcal{L}_{multi-cs} = D_{CS}(f_1, f_2, .., f_C) = -\log \sum_{i=1}^{C-1} \sum_{j>i} \frac{V(f_i, f_j)}{\epsilon \sqrt{V(f_i)V(f_j)}} \tag{6}$$

Here, $\epsilon = \sum_{c=1}^{C-1} c$. Similarly, $D_{CS}(f_1, f_2, .., f_C)$ equals zero if and only if the $C$ distributions $f_1(x), f_2(x), ..., f_C(x)$ are the same. Thus, the objective function for $C$ modalities can be written as:

$$\mathcal{L} = \mathcal{L}_{MSE_1} + ... + \mathcal{L}_{MSE_C} + \alpha\mathcal{L}_{multi-cs_{s_1...s_C}} + \delta\mathcal{L}_{multi-cs_{p_1...p_C}}^{-1} + (\beta\mathcal{L}_{cs_{s_1 p_1}}^{-1} + ... + \beta\mathcal{L}_{cs_{s_C p_C}}^{-1}) \tag{7}$$

Here, $\mathcal{L}_{multi-cs}$ represents the CS loss across different modalities. For the above equation, the number of MSE loss terms and the CS loss terms, which encourage independence between shared and private latent spaces within the same modality, remain consistent with the total number of modalities. To simplify, the weight $\beta$ for promoting independence is set to the same value across different modalities.

## 4 RESULTS
We evaluated our model using three datasets: one simulated dataset and two experimental datasets. All evaluations were conducted on held-out data. The training details are provided in Appendix A.7.

## 4.1 SIMULATED DATASET: 3DSHAPE

We evaluated our model using a simulated dataset consisting of multiple sessions, each containing recordings from two modalities. The "image" modality consists of 3D shapes that varied in orientation, scale, shape, and color, with procedural changes applied to these features over time (Burgess & Kim (2018)). We also simulate the corresponding non-linear time series encoding of the scale and orientation, with temporally-structured noise that periodically changes value between 1 and 4. Further details can be found in Appendix A.2.1.

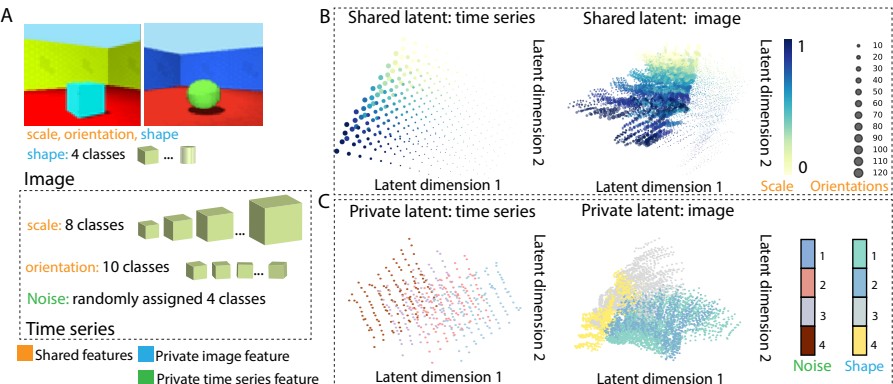

Figure 2: **3Dshape**: A. The image modality consists of 3D shapes with different scales, orientations, and shapes, while the time series modality encodes the scales, orientations, and temporally-structured noise; the shared information across modalities is the scales and orientations. B. Shared latent spaces for each modality, each point represents a single frame, color-coded by orientations and scales. C. Private latent for each modality, color-coded by the corresponding private features: the private latent spaces retains the private features.

Our goal is to capture the shared information present in both image and time series data within the shared latent spaces, while ensuring that the private latent spaces are specific to each modality, retaining only the modality-specific features. By doing so, we can effectively separate the shared time series and image features from those that are unique to either domain. This approach allows for a clearer understanding of the underlying relationships between time series and images, and how the two modalities align or diverge in capturing the intricacies of the data.

To evaluate the success of our model, we compare the correlation matrices across different latent spaces, specifically focusing on the shared and private latents. The shared latent spaces exhibit significantly higher correlation scores compared to the private latent spaces (as detailed in Appendix A.10.1). This higher correlation in the shared spaces suggests that the model effectively captures the common structure between time series and image data, while the private spaces remain distinct, as intended.

We further visualize the latent spaces, with the latents color-coded by orientations and scales (Fig. 2B). The latent spaces show a clear and well-aligned separation between different orientations and scales, further validating the ability of the model to organize key features in the shared latent space. The alignment in the shared space demonstrates that the model was able to identify the common underlying patterns between the two modalities in a way that enhances interpretability.

We apply a linear decoding model to each latent space for feature decoding after binning the data (Fig. 3, Appendix A.12.2). The shared latent spaces demonstrate high accuracy in decoding scales and orientations, highlighting their ability to capture common features across both modalities. The private latent spaces perform well in isolating modality-specific information. For example, the private image latents retain more shape-related details, leading to better decoding accuracy for shape features as compared to the shared latents. Similarly, the private time series latents excel in decoding temporally-structured noise, confirming their success in capturing modality-specific content.

Next, we demonstrate the necessity of applying the CS-divergence to the latent spaces: we train a model using the same architecture but without applying the CS-divergence. As shown in Appendix A.12.2, the results indicate a lack of separation between the private and shared latent spaces, highlighting the necessity of constraints to effectively distinguish shared and private features.

In comparison to previously explored models such as MM-VAE (Shi et al. (2019)) and joint encoding models (Singh Alvarado et al. (2021)), the shared time series and image latents are able to represent the shared features more accurately (Fig. 3, Appendix A.12.2). In comparison models,

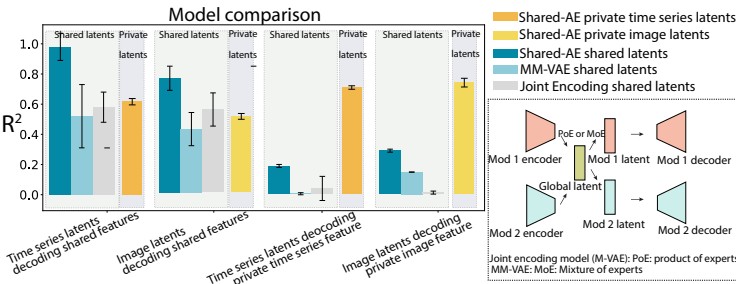

Figure 3: **Comparison against baseline models while having unpaired the data from the modalities**. The Shared-AE shared latent results in better decoding accuracy of shared features, while private latents retain more modality-specific information. The drop in decoding accuracy from the shared latents for baseline models clearly indicates that there is information leakage between modalities during training.

information leakage may result in the blending of modalities, thereby yielding unreliable outcomes in the analysis of relationships between them. For example, in Section 4.2, we aimed to assess the variance explained by each latent subspace. However, if one latent subspace contains features from other modalities (as is prone to happen with the presence of a 'global' latent), it becomes unclear where specific features originate from, leading to ambiguities and confusion in interpreting the latent spaces. Here, we conducted a crucial test to ensure no information leakage between the different modalities. First, we shuffled the time series data while keeping the image data unchanged. We then applied this unpaired data to the trained model to generate new latent representations for both modalities. Using these latent representations, we proceeded with the decoding task. We observed that the decoding results from the latent representation remained unchanged for the unpaired data, confirming that no information leakage occurred between the modalities in Shared-AE (Table 4). This robust performance validates the integrity of the model in maintaining clear boundaries between the shared and private latent spaces, ensuring that shared information is genuinely mutual between the two modalities, and not due to unwanted information transfer or contamination.

In contrast, baseline models demonstrated information leakage, as indicated by a noticeable drop in decoding accuracy when applied to similar tasks (Table 5, Appendix A.12.2). This performance decline highlights the strength of the Shared-AE model in preventing information leakage between modalities, making it a more reliable choice for multi-modal integration. Additionally, the baseline model fails to differentiate between private and shared features, which results in a lack of interpretability in the latent space. Ensuring a clear separation between shared and private latent variables is essential in multi-modal scenarios, especially when handling complex, high-dimensional data. This separation guarantees that the insights derived from the model accurately reflect the true underlying data patterns, rather than being influenced by data contamination. Furthermore, as shown in Appendix A.12.2, when the shared features across both modalities are weak, our model outperforms the rest.

### 4.2 HEAD-FIXED BEHAVIOR: TWO-ALTERNATIVE FORCED CHOICE TASK (2AFC)

We apply Shared-AE to an experimental dataset involving a head-fixed mouse performing a self-initiated visual discrimination task, with behavioral video recordings from two views (face and body), that included the mouse and experimental equipment. We labeled the paws, spouts, and levers using DeepLabCut (DLC). Simultaneously, WFCI across the mouse dorsal cortex was recorded. Further details on the experimental setup are in Appendix A.2.2 and recording / preprocessing details in Musall et al. (2019); Saxena et al. (2020).

**Two Modalities: keypoint positions and WFCI** To explore shared information between pose-estimated behavioral variables and large-scale neural activity (WFCI), we trained Shared-AE with these two modalities. Based on reconstruction results (Appendix A.9.2), we chose a shared latent dimension of 50. Post-training, we extracted latent spaces from the held-out dataset for downstream analyses. We first compared the correlation between the shared latent spaces between the two modal-

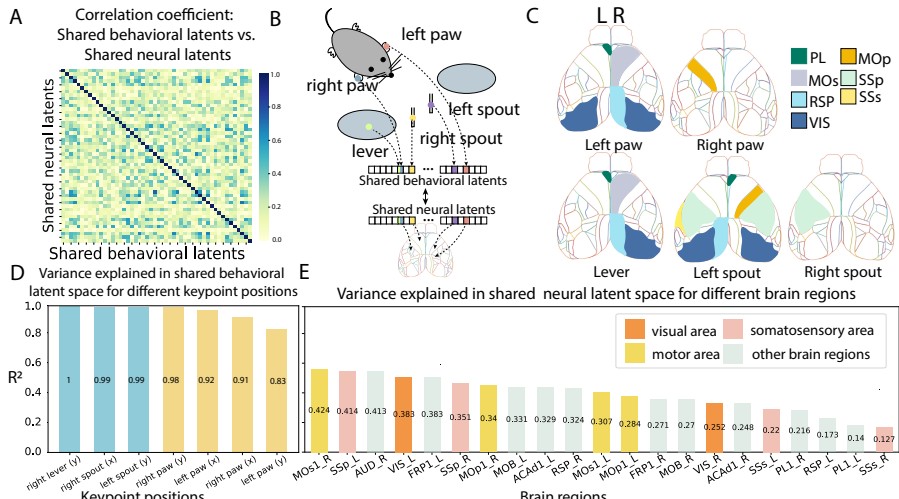

Figure 4: **Head-fixed mouse: visualizing shared latents**: (A) Correlation coefficients between the shared latent spaces: the matrix has a strong diagonal component, indicating that the shared latents are well aligned. (B) In the shared behavioral latent space, we can identify the latent variables that are most helpful in reconstructing each keypoint, and then identify the brain regions that these reconstruct. (C) We identify the regions involved in modulating each keypoint using the procedure in (B). (D)Variance explained for the shared behavioral latents, which indicates how the shared space contributes to the reconstruction of each behavioral variable (equipment in blue and body parts in yellow). E. Variance explained for the shared neural latents, which captures most features from the primary somatosensory and the primary motor area (Abbreviation list in Appendix A.1).

ities. As shown in Fig. 4A, the shared latent spaces of behavioral and neural activities show a high correlation, whereas the private latent spaces do not exhibit this trait (Fig. 9A). Given the high correlation among the shared latents, we next examined the correspondence between the latent spaces and the different features in each modality, i.e., the keypoints and brain regions (Fig.4B). Specifically, we decoded body position and neural activity using each latent in the shared space, establishing a one-to-many correspondence between the latent variables and the underlying data. As illustrated in Fig. 4B, this process allows us to identify the corresponding brain regions for each body position. We see in Fig. 4C the primary brain regions involved with each behavioral variable; here, the contralateral side of the brain is shown to be related to each behavior.

Next, we focus on the role of shared latent variables by isolating them from private latents and employing them to reconstruct the original data using a frozen decoder. This analysis reveals that the equipment in the experiment has considerable shared information with the neural activity since the mouse is performing a task, as compared to the mouse's body parts (Fig. 4D). Notably, the somatosensory and motor cortices have substantial shared information with the behavior (Fig. 4E).

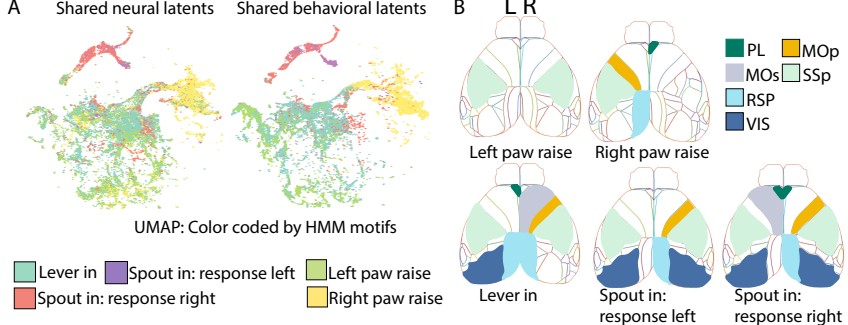

Figure 5: **Head-fixed mouse: shared neuro-behavioral motifs**: A. UMAP representation of the shared neural and behavioral latent subspaces, color-coded by motifs learnt using Hidden Markov Models (HMMs) applied to the shared behavioral subspace; we see that the shared latent successfully captures behavioral motifs, and affords interpretability to the neural subspace.. B. The correspondence between each shared behavioral HMM state and brain regions (details in text).

Finally, we examine whether the shared latents are able to generate reasonable shared neuro-behavioral motifs. To understand this, we apply a Hidden Markov Model (HMM) to the shared behavioral subspace and evaluate its generalization to the shared neural latent in Fig. 5A. In contrast, the private neural latent space was unable to capture the patterns observed in the behavior, highlighting the importance of the shared latent space in identifying cross-modality relationships (Appendix A.13.3). We further calculate the correspondence between various HMM states and brain regions (Fig. 5B): for each HMM state, we compute the variance explained by the shared latent spaces towards each brain region; we only consider those brain regions that are represented with $R^2 > 0.3$. This variance-explained analysis revealed important insights into how different brain regions were involved in distinct behavioral states. For instance, the visual cortex displayed a stronger response to equipment movement, likely reflecting the mouse's reliance on visual cues during task performance. Meanwhile, the somatosensory cortex was associated with nearly all of the identified motifs, suggesting that it played a central role in the task across multiple behaviors.

In comparison with other multi-modal models such as MM-GP-VAE Gondur et al. (2024), PSID Sani et al. (2021), DPAD Sani et al. (2024), MM-VAE Shi et al. (2019), and Joint Encoding Model Singh Alvarado et al. (2021), we emphasize that Shared-AE has better decoding accuracy and offers significantly improved interpretability of the latent space, due to the capability of Shared-AE to form modality-specific shared subspaces. The configuration of each model can be found in Appendix A.13.1. To evaluate decoding accuracy, we trained a linear regression model on the training dataset to predict body position using the neural latent representations. In cases where a decoder was readily available during training, such as in PSID and DPAD, we directly used it to generate these predictions (Table 1, Appendix A.13.2). Additionally, we compared our model with the baseline models on unpaired datasets, where Shared-AE has better decoding accuracy (Appendix A.13.2).

Table 1: Behavioral decoding accuracy with paired modalities for 2AFC body positions dataset

| Subspace | PSID | DPAD | MM-VAE | MM-GP-VAE | **Shared-AE** |
|---|---|---|---|---|---|
| Private latents | $NA$ | $NA$ | $NA$ | $0.37 \pm 0.00$ | $0.22 \pm 0.03$ |
| Shared latents | $0.20 \pm 0.03$ | $0.27 \pm 0.02$ | $0.41 \pm 0.07$ | $0.36 \pm 0.01$ | $0.41 \pm 0.05$ |

**Two Modalities: behavioral video and neural activity** To test the generalization of our model, we apply Shared-AE to the same head-fixed dataset, while changing the behavioral modality from keypoints to high-dimensional behavioral images. We see a high correlation between the shared behavioral and neural latents (Appendix 9C). Using latent traversals, which systematically vary the latent variables to reveal their influence on the model's output and help identify specific behavioral features encoded in the latent space, we demonstrate that the shared behavioral latent captures the movements of the paw, spout, and lever (Fig. 15A). The private latent however captures the appearance of the mouse such as the shape of the eyes (Appendix A.13.4). Additionally, we compare the neural encoding results using shared behavioral latents with (i) the behavioral videos, and (ii) the keypoint positions being the behavioral modality. We found that during task-unrelated behaviors, such as raising the left and right paws, the images' shared behavioral latents show significantly higher accuracy for encoding the visual and motor areas (Fig. 15B, Appendix A.11). The behavioral image captures more features than just keypoint positions, which is consistent with the findings reported in the original paper. (Musall et al. (2019)). Moreover, unlike models such as PSID, Shared-AE can handle image data, providing greater flexibility for complex multimodal tasks. As shown in Appendix A.13.2, Shared-AE outperformed the baseline models.

**Three Modalities: body position, behavioral video, and neural activity** Shared-AE can be extended to more than two modalities; here, we show this capability using body positions, behavioral videos, and neural activity from the head-fixed mouse dataset. The shared latent spaces of these three modalities are highly correlated (Appendix 9B). To test whether the shared neural latents capture details about the behavior, we predicted future keypoint positions and compared this to predictions using shared neural latents from two modalities. Results show higher accuracy in predicting right paw movement with the three-modality model, especially for potentially task-unrelated states such as 'right paw raise', while task-related predictions remain comparable to the two-modality model.

### 4.3 SOCIAL BEHAVIOR

To further evaluate the performance of our model in a more complex behavioral setting, we conducted experiments on a social behavior dataset. In this scenario, two mice, referred to as the agent (m1) and subject (m2), were engaged in social interactions, while mesoscopic imaging was simultaneously performed using a large field-of-view miniscope on the subject. For more information, see

Appendix A.2.3. The body positions of the two mice were labeled by DLC (Lauer et al. (2021)), while the neural activity was preprocessed by LocaNMF (Saxena et al. (2020)). In addition to tracking the body positions, we extracted several socially-relevant behavioral features, such as the nose-to-nose distance between the two mice, the relative angle between their body orientations, and other key interaction metrics. These behavioral features allowed us to quantify the complexity of social behaviors and understand their relationships with neural activity.

Based on the reconstruction accuracy (Appendix A.9.3), we set the latent dimension to be 60. We apply principal component analysis (PCA) to the shared latent spaces and show high correlations across the dominant PCs of the different shared latent subspaces (Fig. 6A). The private latent spaces do not exhibit this trait (Appendix A.10.3). The latents effectively capture the temporal features, and the neural and behavioral latents are well aligned (Fig. 6B).

Next, we perform latent traversals to visualize the contribution of each latent space. As expected, as shown in Fig. 6C, the shared neural latents capture a higher $R^2$ for the subject's behavior as compared to the agent's. This supports the idea that neural encoding primarily reflects the subject's perspective and interaction within the environment. However, the shared behavioral latents also capture the agent's behavior and many social features, such as the nose-to-tail distance, suggesting that proximity to the agent and the agent's behavior itself plays a significant role in modulating neural activity. Furthermore, the shared neural latents show widespread activation across regions, including the somatosensory and motor areas (Fig. 6D). These areas are critical for movement and sensory integration, further validating the model's capacity to identify relevant neural substrates that underpin behavior. This alignment between neural activity and behavior underscores the robustness of the shared latent representations in capturing social dynamics in multi-agent settings.

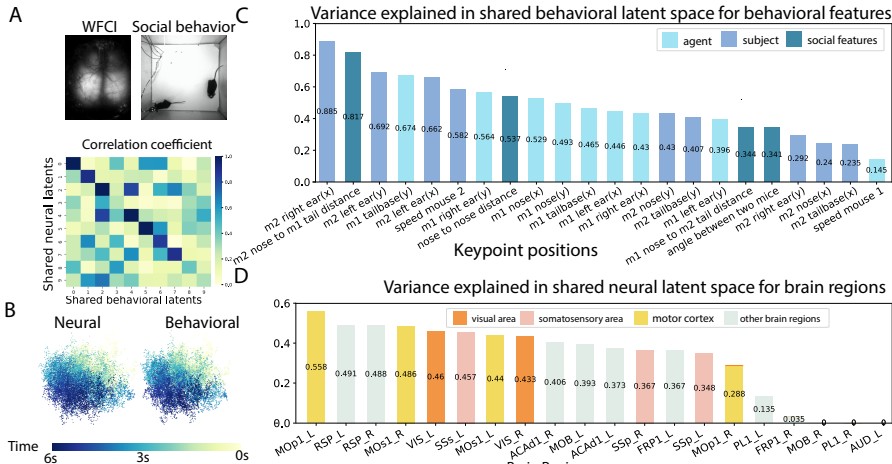

Figure 6: **Socially behaving mice**: A. Example neural and behavioral data (top); correlation between the shared latent spaces (bottom). B. Lower dimensional embedding for each modality found using PCA, color-coded by the time spent in the arena. C-D: Variance explained by each shared latent subspace: the shared behavioral latents capture features related to subject and agent, as well as some social features, represented in the shared neural subspace with distributed neural coding.

## 5 CONCLUSION AND LIMITATIONS

This paper introduces Shared-AE, an AE-based framework using CS regularization to identify features common to both behavior and neural activity, especially useful in settings where both modalities are high-dimensional and represent complex behavior. By utilizing CS divergence and its inverse, Shared-AE captures both shared and unique features across modalities, enhancing our understanding of the relationship between neural activity and behavior. Despite its numerous hyperparameters, the model remains robust when well-trained. Limitations include its requirement for equal latent subspace (Appendix A.5) dimensions, which can be inflexible, and its generalizability across multiple subjects is uncertain. Overall, Shared-AE is a robust tool for multi-modal research. Future work will explore more complex encoder models and pre-trained networks to improve training efficiency and the ability to capture informative features. Here, we aim to achieve neuroscientific insight, and do not note any negative societal impact.

ACKNOWLEDGMENTS

We gratefully acknowledge support from the NIH Brain Initiative 1R34DA059718-01 and NSF NCS 2350329. We thank Nancy Padilla-Coreano for helpful discussions.

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

# A  APPENDIX

## A.1  ABBREVIATION LIST

A detailed abbreviation list for different brain regions can be found at Allen Institute for Brain Science

## A.2  DATASET

### A.2.1  DATASET: 3DSHAPE

The 3DShapes dataset is composed of procedurally generated 3D objects, each defined by six independent ground truth latent factors: floor color, wall color, object color, scale, shape, and orientation. Every possible combination of these factors is uniquely represented, resulting in a dataset of 480,000 images. For our task, we converted the images to grayscale to remove the color-related features, leaving three key attributes: (1) scale, with 8 values linearly spaced between 0 and 1, (2) shape, with 4 distinct categories [0, 1, 2, 3], and (3) orientation, with 15 values linearly spaced between -30 and 30 degrees. For the labels, we retained the scale and orientation values and introduced a temporally-structured noise that periodically changes between one and four. For time efficiency, we applied 8000 images for training and 2000 images for testing.

### A.2.2  DATASET: TWO-ALTERNATIVE FORCED CHOICE TASK (2AFC)

We employed a subset of the behavioral dataset from Musall et al. (2019). The task involved mice pressing a lever to start, displaying a visual stimulus to the left or right, and then making a decision by licking a spout corresponding to the stimulus direction after a delay. Correct choices were rewarded with juice. Behavior and neural activity were recorded at 30 Hz. The training set consists of 388 trials, and the test set contains 128 trials, each comprising 189 frames.

### A.2.3  DATASET: SOCIAL BEHAVIOR

Our study used a social behavior dataset involving simultaneous brain mesoscopic imaging and natural behavior recording in mice. Mesoscopic imaging was performed using a large field-of-view miniscope, while behavior was captured by three cameras at different angles. Two GCaMP6-expressing mice (slc17a7-cre x ai162) were observed in a cubic box arena, with one mouse equipped with a miniscope. The video, around 18 minutes long, was split into 326 chunks, with a 3:1 training to test data ratio. Social features such as nose-to-nose distance, tail-to-nose distance, speed, and angle between mice were calculated, resulting in 37 features in the behavioral datasets.

## A.3  TECHNICAL AND SCIENTIFIC NOVELTY OF SHARED-AE

Shared-AE introduces a novel approach to joint neural and behavioral modeling, addressing key limitations in existing methods and offering several distinct advantages:

**(a) Enhanced Interpretability through Latent Subspace Separation:** Unlike the model by Gondur et al. (2024), which combines neural and behavioral latents into a single subspace, Shared-AE explicitly separates shared and private latent spaces. This separation preserves interpretability by reducing information leakage across modalities, allowing us to better disentangle distinct modality-specific features. Indeed, having common shared subspaces can be detrimental towards interpretability, since it is unclear which modality is leading to the performance of the shared latent. As an additional key consequence of modality-specific shared subspaces, if we only have data from one modality during inference, we do not require data from the other modalities to generate robust and meaningful representations.

**(b) Improved Performance on Paired and Unpaired Tasks:** We designed an 'unpaired' task to ensure there is no information leakage between different modalities. Critically, after training the model, if one modality's data is corrupted during inference, this should not affect the other modality's latent representation or decodability. Here, during inference only, one modality was shuffled across time, while the other modality remained unchanged. The resulting 'unpaired' data was then input into the trained model to generate new latent representations for the unshuffled modality. This approach allowed us to assess whether the model could maintain the integrity of the

unshuffled modality's latent features, ensuring that the latent representations were unaffected by the shuffled modality. We compared Shared-AE against existing approaches on the 2AFC dataset (as requested by the reviewers), demonstrating superior performance in both paired and unpaired tasks (Table 11, 12). Shared-AE outperforms other models by effectively capturing complex relationships across modalities, and preserves information in one modality even when the other modality may be corrupted during inference. This robustness is crucial for practical applications where complete data may not always be available.

**(c) Flexibility with Multiple Modalities and Image Data:** Shared-AE is designed to handle more than three modalities, including image data, significantly broadening its application scope from previous research; in fact, this capability has not previously been shown in any multi-modal neuroscience study. As shown in Figure 6B, using raw behavioral images as well as pose estimation leads to better prediction accuracy of neural activity compared to using pose estimation alone. This finding aligns with Musall et al. (2019), indicating that raw image data provides a richer representation of behavior. In contrast, existing models such as PSID are limited in processing data from more than two modalities effectively.

**(d) Minimizing Distribution Distance Instead of Predefined Priors:** Previous works such as Yi et al. (2022) and Tran et al. (2021) do indeed use a CS-divergence, but in a drastically different way than in our study: they use CS-divergence to fit the latent distribution to a *predefined prior*, whereas Shared-AE minimizes the distance between two *learned* distributions instead. This approach avoids the limitations of predefined priors and allows for more flexible and meaningful representations.

**(e) Utility in Downstream Tasks and Enhanced Variance Explained:** By separating the latent features into distinct shared and private subspaces, Shared-AE provides representations that can be effectively used for multiple downstream tasks.

## A.4 CS INEQUALITY

For two functions $h(x)$ and $g(x)$, the Cauchy-Schwarz inequality is expressed as:

$$\left| \int h(x)g(x)dx \right|^2 \leq \int |h(x)|^2 dx \int |g(x)|^2 dx \tag{8}$$

with equality holding if and only if the two functions are linearly dependent.

## A.5 EQUAL LATENT SUBSPACE DIMENSIONS

Mathematically, the CS-divergence requires both distributions to have the same dimensionality, allowing for the calculation of cosine similarity after kernelizing the latent representations. In practice, as illustrated in Fig. 2, the dimensionality of the latent space often exceeds the actual number of features. This implies that when the latent dimension is large, there may be redundancy in each latent subspace. To overcome this limitation in practice, we perform dimensionality reduction on the latent subspaces after training, especially for visualization purposes.

## A.6 RELATIONSHIP BETWEEN VAE AND THE CS REGULARIZED AE

For a standard VAE with a Gaussian prior, the Evidence Lower Bound (ELBO) is defined as:

$$
\begin{aligned}
ELBO &= \mathbb{E}_{\hat{z} \sim q_\phi(z|x)}[\log p_\theta(x|\hat{z}) + \log p_\theta(\hat{z})] + \mathbb{H}[q_\phi(\hat{z}|x)] && (9) \\
&= \mathbb{E}_{\hat{z} \sim q_\phi(z|x)}[\log p_\theta(x|\hat{z})] + \mathbb{E}_{\hat{z} \sim q_\phi(z|x)}[\log p_\theta(\hat{z})] + \mathbb{H}[q_\phi(\hat{z}|x)] && (10) \\
&= \mathbb{E}_{\hat{z} \sim q_\phi(z|x)}[\log p_\theta(x|\hat{z})] - D_{KL}(q_\phi(\hat{z}|x)||p_\theta(z)) && (11)
\end{aligned}
$$

Where $x$ is the input, $z \sim \mathcal{N}(0,1)$, $\hat{z}$ is the learned latent, and $p_\theta(x|\hat{z}) = \mathcal{N}(\mu_{nn}(z), \sigma_{nn}(z))$. During training, the goal is to maximize the ELBO. Therefore, the objective function is written as:

$$\mathcal{L}_\mathcal{N} = D_{KL}(q_\phi(\hat{z}|x)||p(z)) - \mathbb{E}_{\hat{z} \sim q(z|x)}[\log p_\theta(x|\hat{z})] \tag{12}$$

In contrast, our model, Shared-AE, utilizes the CS-divergence for regularization. Unlike the VAE, which employs KL divergence to measure the difference between the approximate posterior and

the prior, Shared-AE leverages the CS divergence to encourage the alignment of the approximate posterior with the prior, enhancing flexibility in capturing the underlying structure of the data.

To place Shared-AE in a probabilistic framework, we consider the objective to maximize the log marginal likelihood of the model, as below (Tran et al. (2021)).

$$\max_\theta \mathbb{E}[\log p_\theta(x)] = \max_\theta \mathbb{E}_{p(x)}\left[\log \mathbb{E}_{p(z)}[p_\theta(x|\hat{z})]\right], \tag{13}$$

Using Jensen's inequality, we can obtain a lower bound to the log-marginal likelihood as follows:

$$\log p_\theta(x) = \log \mathbb{E}_{\hat{z}\sim p(z)}[p_\theta(x|\hat{z})] \geq \mathbb{E}_{\hat{z}\sim p(z)}[\log p_\theta(x|\hat{z})]. \tag{14}$$

Similarly, as in the VAE framework, we define a mapping $q_\phi(z|x)$ which transforms some input $x$ to (probabilistic) features $z$. By adding a regularization $R$, we penalize any deviation between $q_\phi(z|x)$ from $p(z)$. Ideally, this regularization is a metric function for which $R > 0$ when $q \neq p$ and $R = 0$ if and only if $q = p$.

$$\max_{\theta,\phi} \mathbb{E}_{p(x)}\mathbb{E}_{\hat{z}\sim q_\phi(z|x)}[\log p_\theta(x|\hat{z})] \tag{15}$$

$$\text{subject to } 0 \leq R(q_\phi) < \epsilon. \tag{16}$$

In this formulation, $\epsilon$ specifies the magnitude of the applied constraint. If $R$ is defined as the KL divergence, we have the original ELBO formulation. We diverge from this principle and use the Cauchy-Schwarz divergence for regularization to match an approximate posterior to a prior, with the advantage of added flexibility and expressiveness. Thus, the objective function is given by:

$$\max_{\theta,\phi} \mathbb{E}_{p(x)}\left[\mathbb{E}_{\hat{z}\sim q_\phi(z|x)}[\log p_\theta(x|\hat{z})]\right] \tag{17}$$

$$\text{subject to } D_{CS}(q_\phi(\hat{z}|x)\|p(z)) < \epsilon. \tag{18}$$

Rewriting this as a Lagrangian, we obtain:

$$\mathcal{F}(x;\theta,\phi,\lambda) = \mathbb{E}_{\hat{z}\sim q_\phi(z|x)}[\log p_\theta(x|\hat{z})] - \lambda(D_{CS}(q_\phi(\hat{z}|x)\|p(z)) - \epsilon), \tag{19}$$

where $\lambda$ is the regularization coefficient ensuring that the posterior distribution is close to the prior $p(z)$. We can rewrite this as:

$$\mathcal{F}(x;\theta,\phi,\lambda) \geq \mathbb{E}_{\hat{z}\sim q_\phi(z|x)}[\log p_\theta(x|\hat{z})] - \lambda D_{CS}(q_\phi(\hat{z}|x)\|p(z)) =: L_{CS-AE}(x;\theta,\phi,\lambda). \tag{20}$$

While VAEs rely on KL divergence to regularize the latent space and align the approximate posterior with the Gaussian prior, CS-AE uses CS divergence, allowing for more flexible and potentially richer latent space representations. Moreover, our formulation in Shared-AE further defines a specific set of latent variables from one modality to regularize using the latent variables obtained from a different modality, to elucidate the shared structure in the data.

## A.7 TRAINING DETAILS

All models were trained and tested on a single NVIDIA A100 using PyTorch 2.0.1. For our dataset, the runtime per batch varied based on the modalities used: approximately 0.398 seconds for keypoint positions and neural activity, 4.523 seconds for behavioral images and neural activity, and 5.575 seconds for keypoint positions, behavioral images, and neural activity. Each model was trained for 100 epochs with a batch size of 256, using the Adam optimizer with a learning rate of $1e - 4$.

### A.7.1 3DSHAPE

For each input, we applied a window size of one. A 2D ResNet-18 backbone was used for image input, and a 1D ResNet-18 backbone for time series data. A 2-layer 2D convolutional decoder was applied to image data and a 2-layer 1D convolutional decoder to time series data. For classification tasks, scale and orientation are the shared features between image and time series data. The shape is the image-only feature while the temporally-structured noise is the time series-only feature.

### A.7.2 2AFC

A sequence of 9 frames was stacked together, based on the HMM output of keypoint positions. Each sequence spans approximately 0.3 seconds. For keypoint positions and neural activity, a 1D ResNet-18 backbone was used for encoding, and a 4-layer 2D convolutional decoder for behavioral images and a 2-layer 1D convolutional decoder for keypoint positions and neural activity.

### A.7.3   SOCIAL BEHAVIOR

A sequence of 8 frames was stacked together based on the HMM output of keypoint positions. A 1D ResNet-18 encoder and a 2-layer 1D convolutional decoder were used for both modalities.

### A.8   STATISTIC TESTS

We performed t-tests on the test dataset and used the p-value to determine the significance of our results. The p-value annotation legend is:

$$
\begin{cases}
\text{ns:} & 5.00 \times 10^{-2} < p \le 1.00 \times 10^{0} \\
\text{*:} & 1.00 \times 10^{-2} < p \le 5.00 \times 10^{-2} \\
\text{**:} & 1.00 \times 10^{-3} < p \le 1.00 \times 10^{-2} \\
\text{***:} & 1.00 \times 10^{-4} < p \le 1.00 \times 10^{-3} \\
\text{****:} & p \le 1.00 \times 10^{-4}
\end{cases}
$$

### A.9   HYPERPARAMETER SEARCH

This section describes the procedures for selecting parameters. Due to the nature of the loss, each subspace should have the same number of latent dimensions.

### A.9.1   3DSHAPE

The original dataset includes 6 different features. To simplify the dataset, we grayscaled the images to have only 3 features. For better reconstruction accuracy, we choose the latent dimensions equal to 5. The mean MSE loss for the image and the mean R2 for the time series are 0.23e 5 and $0.98 \pm 0.007$, respectively. The kernel size is set to 15 and the weight for all the CS and inverse loss terms is 10.

The final training CS loss for the shared latent space is 0.0009; the inverse CS loss is $9.78 \pm 0.03$ for the individual shared and private latent spaces; the inverse CS loss for the private latent space of different modalities is 9.78.

### A.9.2   2AFC

We choose the latent dimension based on the reconstruction accuracy using 5-fold cross-validation: the smallest value when the MSE loss converged. For keypoint positions, 7 different parts are encountered, and we show the mean and maximum $R^2$ values for all keypoints. Similarly, for neural activity with 21 different regions, we show the mean and maximum $R^2$ values across all brain regions (Figure 7). We evaluated the decoding accuracy of body positions using shared neural latents with varying latent dimensions: 50, 80, and 100. As shown in Table 2, the decoding accuracy remains relatively consistent across these dimensions, indicating that the model's performance is robust to changes in latent dimensionality. For this dataset, we chose a latent dimension equal to 50. The held-out data served as the test set on which all the results are reported. For a two-modality

Table 2: Behavioral decoding accuracy with various latent dimensions using Shared-AE for 2AFC dataset

|  | 50 | 80 | 100 |
|---|---|---|---|
| Shared neural latent | $0.41 \pm 0.05$ | $0.43 \pm 0.03$ | $0.42 \pm 0.04$ |

task with the image as the behavioral input, the reconstruction result for the image is evaluated by $MSE$. We choose 85 as the latent dimension. For the three modalities tasks, we choose the latent dimension to be 85 for comparison. The mean MSE loss and the mean $R^2$ are comparable to the two modality results being $0.1e - 5$ and $0.65 \pm 0.2$, respectively.

For other hyperparameters, we set the kernel size $\sigma$ for this dataset to be 15 (Yi et al. (2022)). For simplification, we set $\alpha, \beta, \gamma,$ and $\theta$ to be the same, all of them equal 5. The results are robust to changes in these hyperparameters, as long as the losses converge to a certain range. In Table 3, we

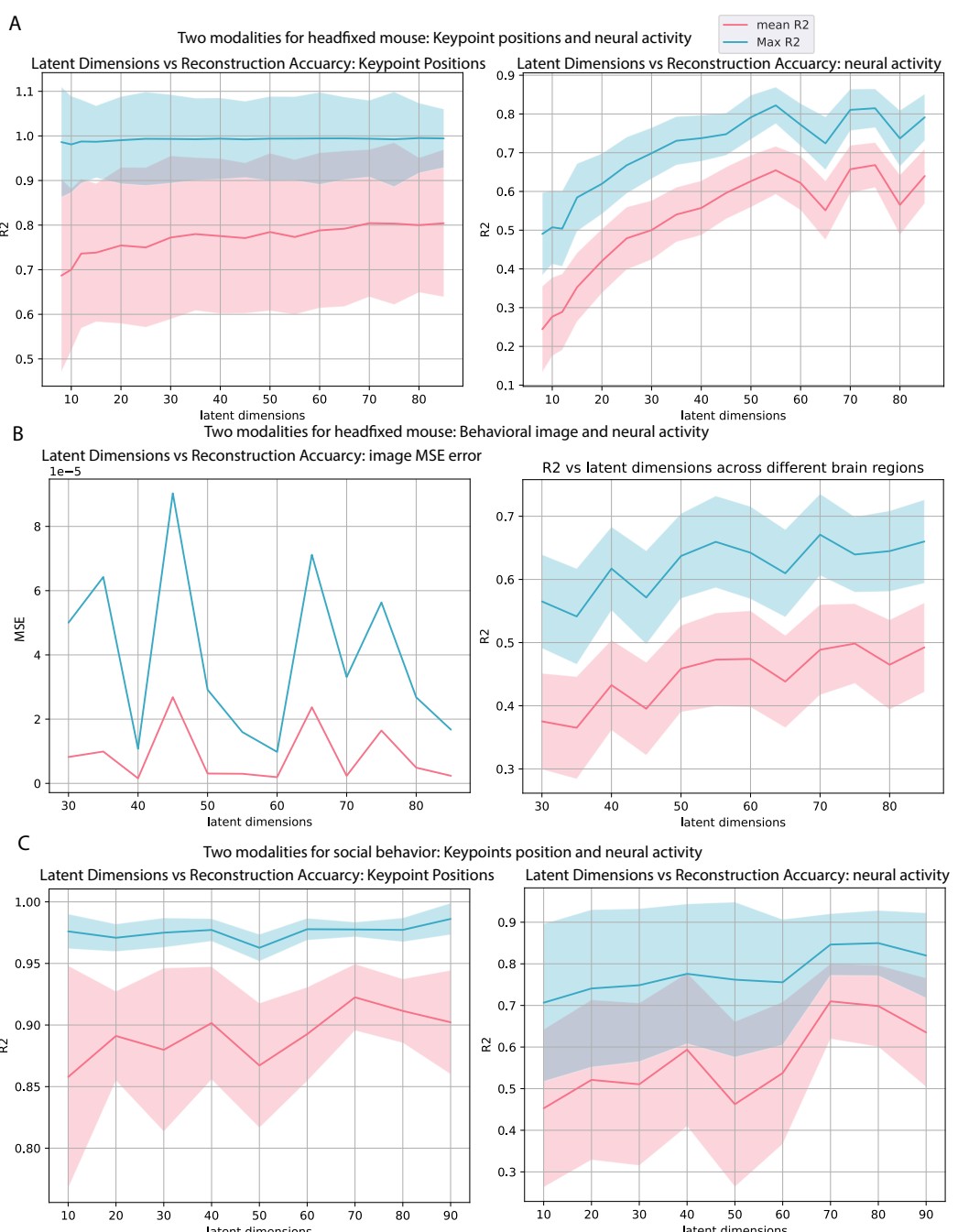

Figure 7: Reconstruction accuracy for different datasets with different latent dimensions. The mean R² represents the average accuracy across all keypoint positions and neural channels, while the max R² indicates the maximum accuracy observed among these keypoints and channels. This distinction helps capture both the overall performance and the peak accuracy of our model across different features, providing a more comprehensive evaluation of its predictive capabilities: A. 2AFC: 2 modal tasks with keypoint positions and neural activity B. 2AFC: 2 modal tasks with behavioral image and neural activity C. social behavior: 2 modal tasks with keypoint positions and neural activity

include the after-training CS divergence loss for each task. For simplicity, we calculated the mean value of the inverse CS loss for the same modality (reported before inversion).

Table 3: CS loss and inverse CS loss values for head fixed dataset

| tasks | cs loss for shared latents | inverse cs loss for the same modal | inverse cs loss for different modal |
|---|---|---|---|
| keypoint positions+ neural activity | 0.01 | $8.65 \pm 0.05$ | 6.77 |
| image+ neural activity | 0.019 | $8.59 \pm 0.007$ | 5.16 |

### A.9.3 SOCIAL BEHAVIOR

According to the reconstruction plot in Fig.7C, we chose 60 as the number of latent dimensions. The kernel size is set to 15 and the weight for all the CS and inverse loss terms is 10. The final training CS loss for the shared latent space is 0.006; the inverse CS loss is $8.35 \pm 0.3$ for the individual shared and private latent spaces; the inverse CS loss for the private latent space of different modalities is 6.99.

### A.10 CORRELATION COEFFICIENT FOR DIFFERENT LATENT SUBSPACES

### A.10.1 3DSHAPE

The correlation coefficient matrices for different latent subspaces are shown in Fig. 8. Despite some correlations between the shared and private image latent spaces, the other subspaces are well separated.

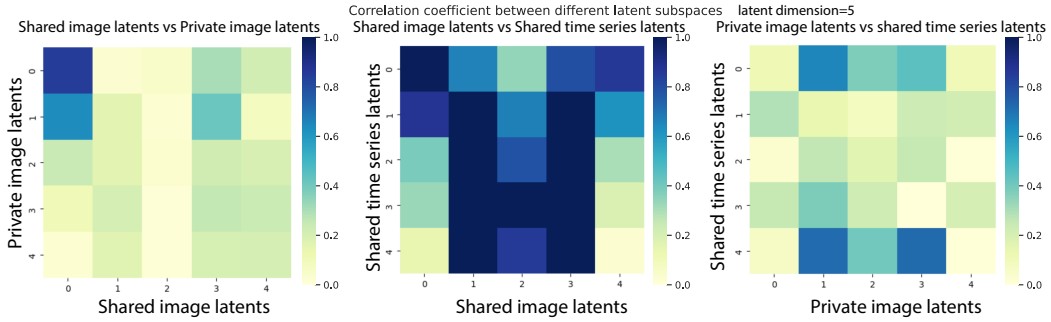

Figure 8: 3Dshape: Correlation coefficient for different latent subspaces.

### A.10.2 2AFC

The correlation coefficient matrices for different latent subspaces are shown in Fig. 9. As expected, the shared latents exhibit high correlations, while the shared and private latents for the same modality show lower correlations.

### A.10.3 SOCIAL BEHAVIOR

The correlation coefficient matrices for different latent subspaces are shown in Fig. 10. Despite some correlations between the shared and private social latent spaces, the other subspaces are well separated.

### A.11 PREDICTION TASKS FOR DIFFERENT MODALITIES

We compared the prediction accuracy between image behavioral latents and keypoint positions behavioral latents for different HMM states (Figs. 11 and 12).

We applied the shared neural latent generated by the three-modal tasks for behavior prediction (Fig. 13).

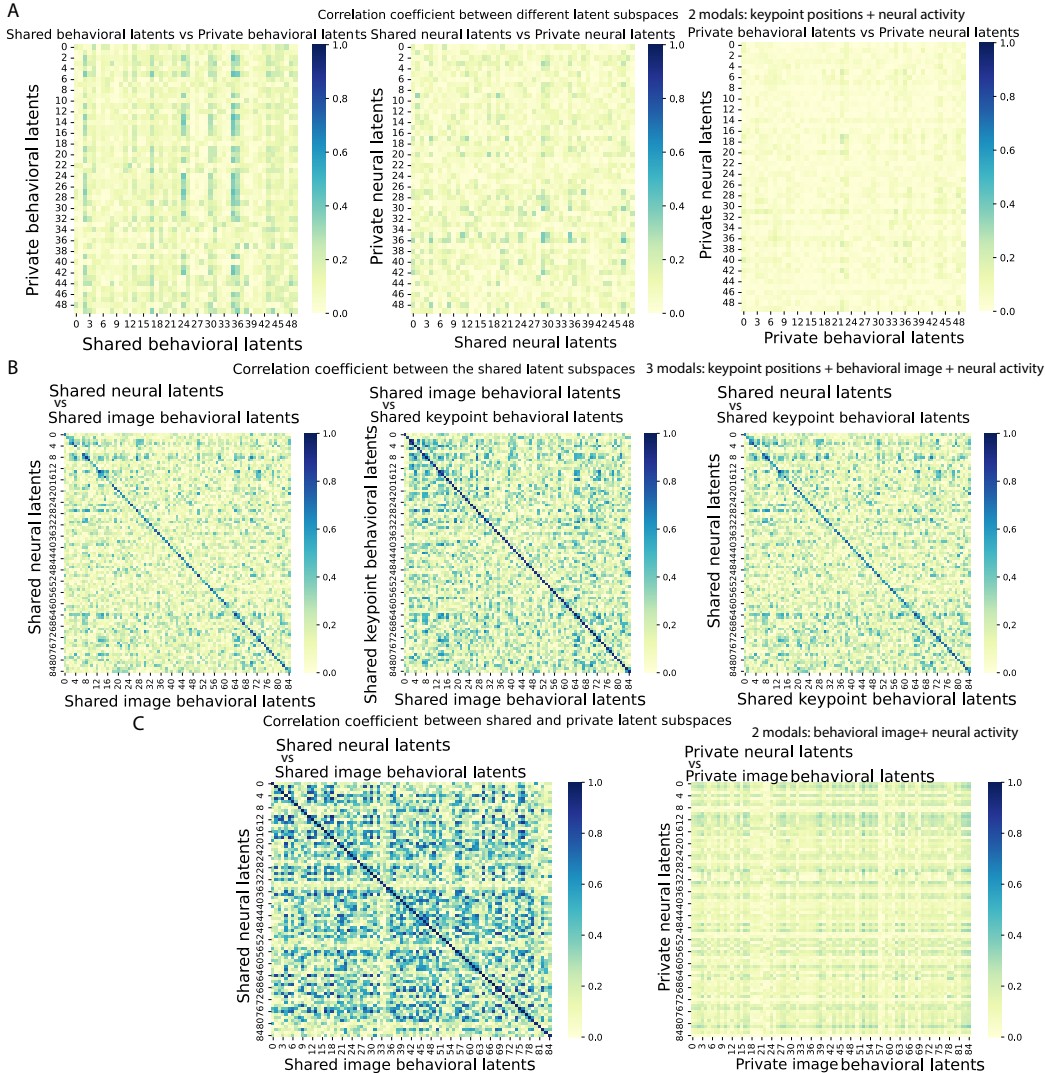

Figure 9: 2AFC: Correlation coefficient for different latent subspaces. A. Comparison between different latent spaces for the 2-modal task: keypoint positions and neural activity. (Correlation should be low for shared vs. private and private vs. private) B. Comparison between shared latent spaces for the 2-modal task: behavioral image and neural activity. C. Comparison between different latent spaces for the 3-modal task: behavioral image, keypoint positions, and neural activity.

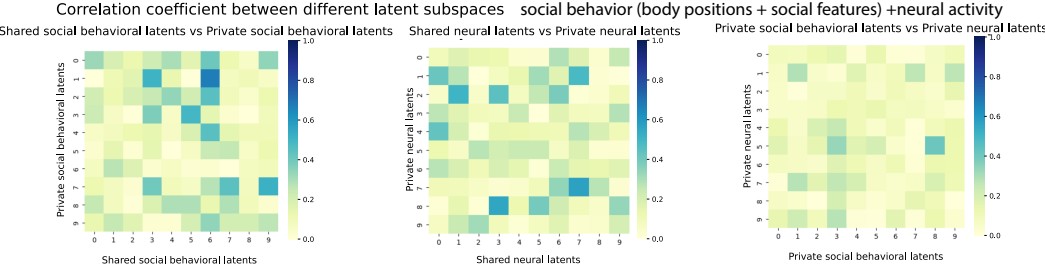

Figure 10: Social behavior: Correlation coefficient for different latent subspaces.

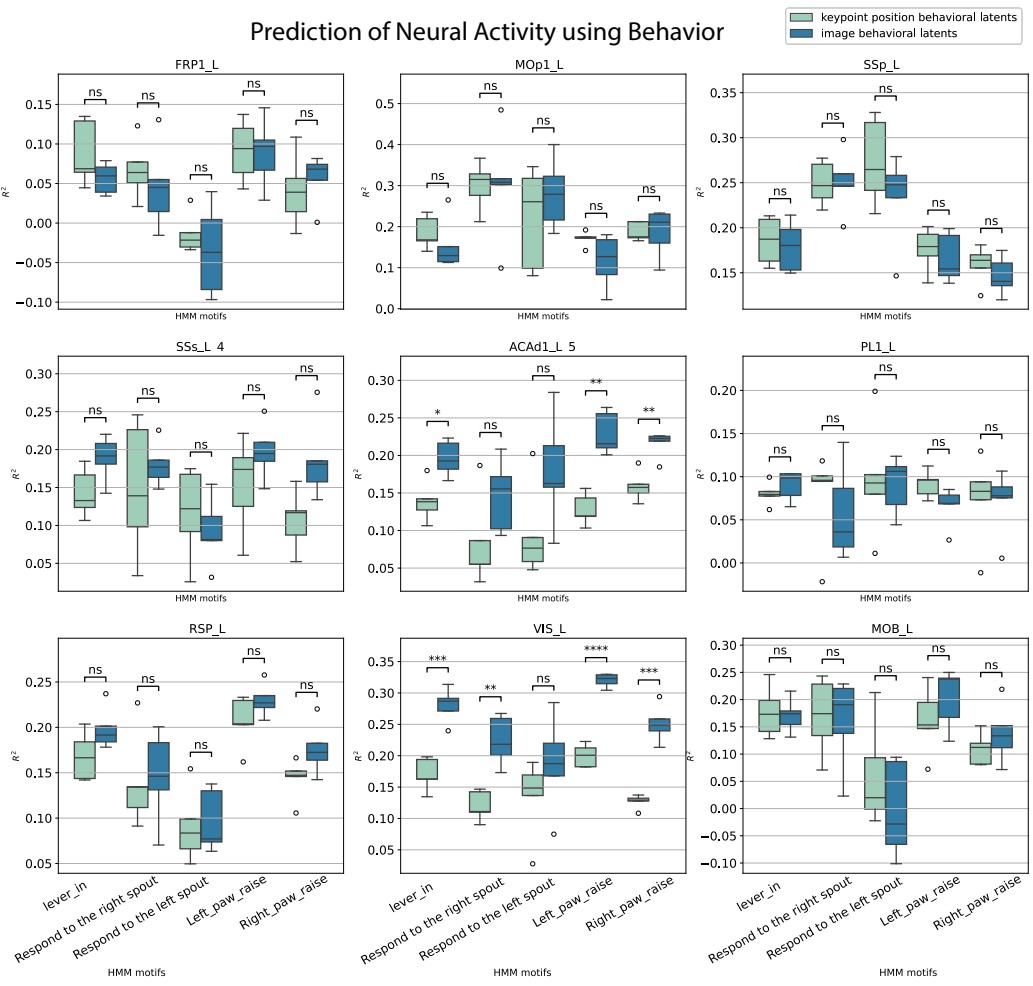

Figure 11: Prediction accuracy of neural activity (A): comparison between the image behavioral latents and key point positions behavioral latents.

## A.12    ADDITIONAL RESULTS: **3Dshape dataset**

### A.12.1    PCA EMBEDDINGS OF THE SHARED AND PRIVATE LATENT SUBSPACE

Fig. 14 illustrates that the shared latent does not capture private features, while the private latent exhibits lower decoding accuracy for shared features. This further demonstrates that our model effectively generates distinct shared and private latents.

### A.12.2    BASELINE COMPARISON WITH PSID AND DPAD

The key advantage of the Shared-AE model over other approaches lies in its ability to effectively disentangle shared and private features, resulting in a clearer separation of modality-specific information. This distinct separation not only enhances the interpretability of the latent representations but also supports a wider range of downstream tasks across different data modalities. As shown in Table 6, when the shared features across both modalities are weak, both PSID and DPAD struggle to effectively separate the private features from the shared latent spaces. Here, both models resulted in an erroneously high decoding accuracy of a private feature from the shared latent space. Shared-AE was able to successfully decouple the shared vs. private features in its latent subspaces.

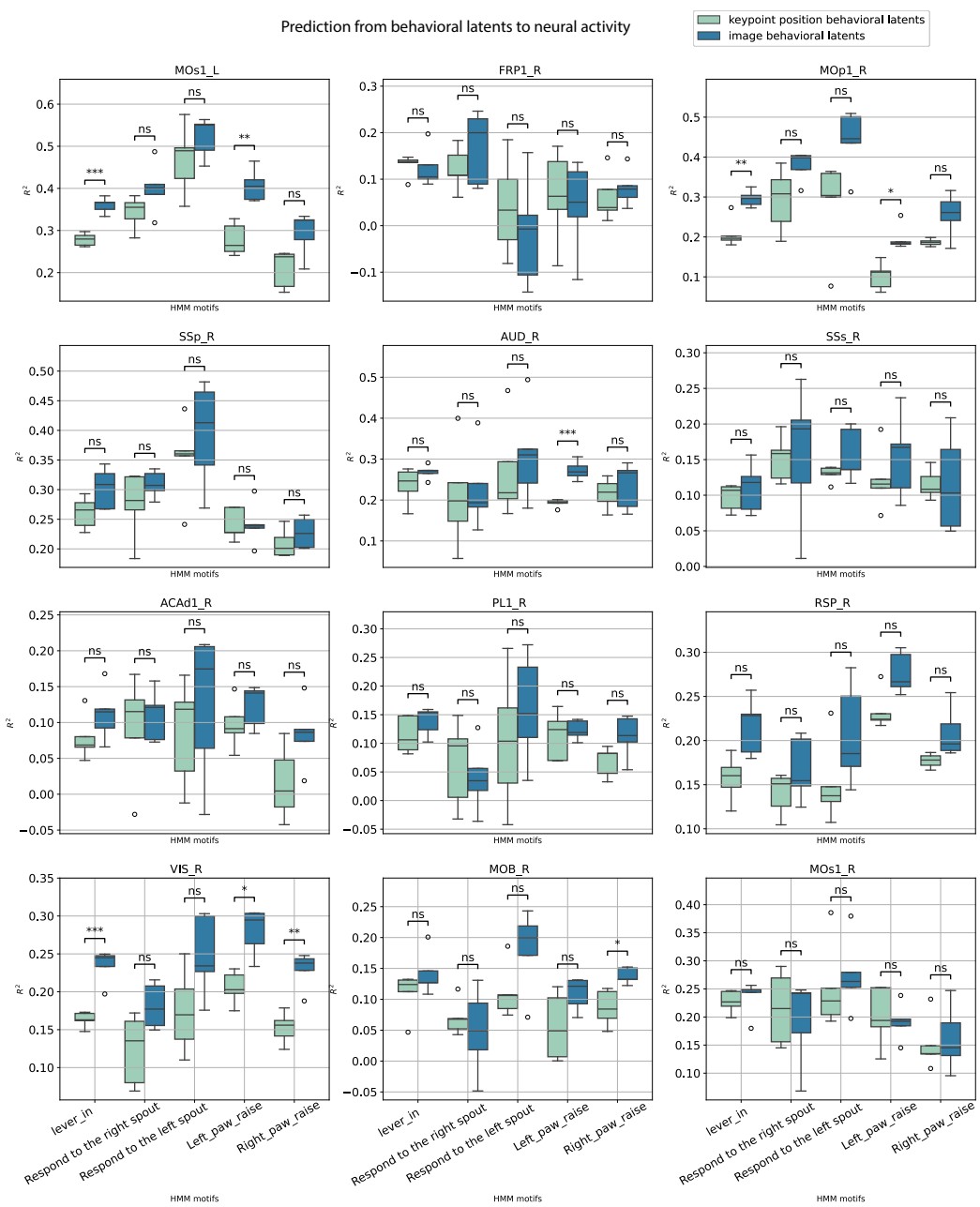

Figure 12: Prediction accuracy of neural activity (B): comparison between the image behavioral latents and key point positions behavioral latents

### A.12.3 BASELINE COMPARISON WITH BASELINE MODEL WITHOUT CS-DIVERGENCE

As a baseline, we trained the model on a simulated dataset using the same architecture but without applying CS-divergence. As shown in Table 7, the results indicate a lack of separation between the private and shared latent spaces, highlighting the necessity of constraints to effectively distinguish shared and private features. Importantly, the shared subspaces in the Shared-AE are not in fact reconstructing both modalities; they remain completely separate from each other, with purely the CS-divergence to regularize them to be similar.

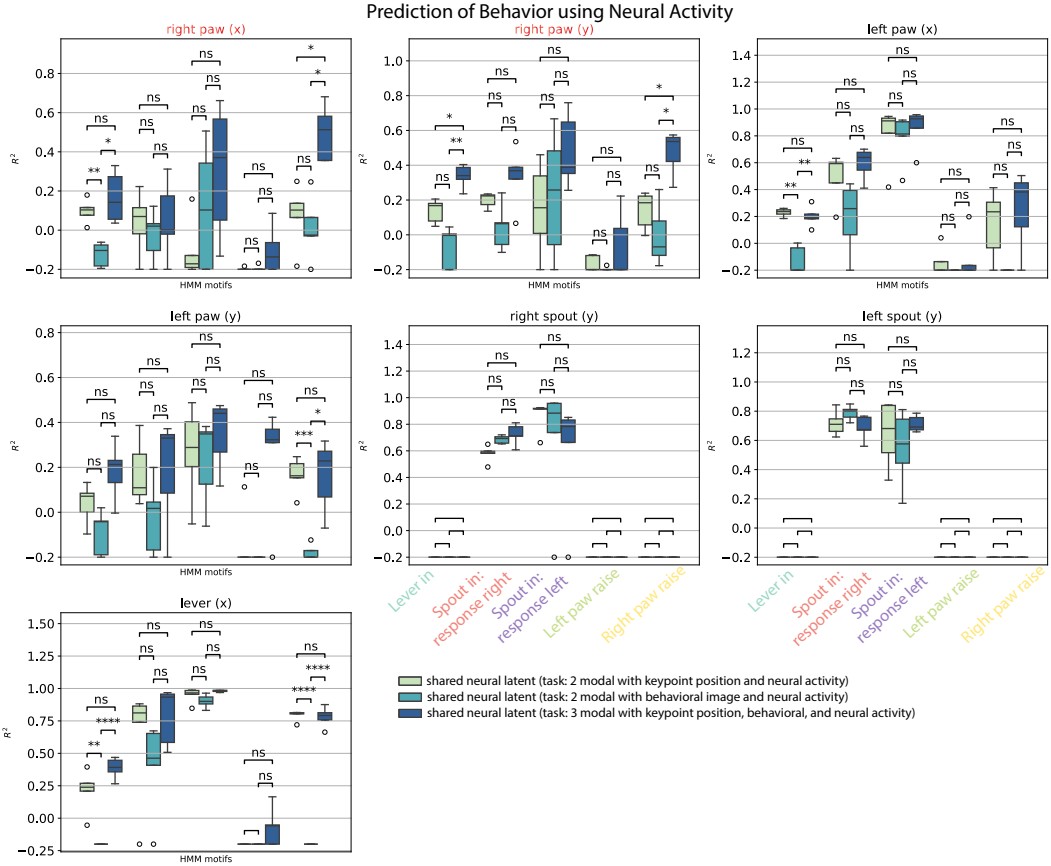

Figure 13: Prediction of keypoint positions using shared neural latents shows that incorporating the third modality (behavioral image) enhances the neural latents' ability to capture intricate features.

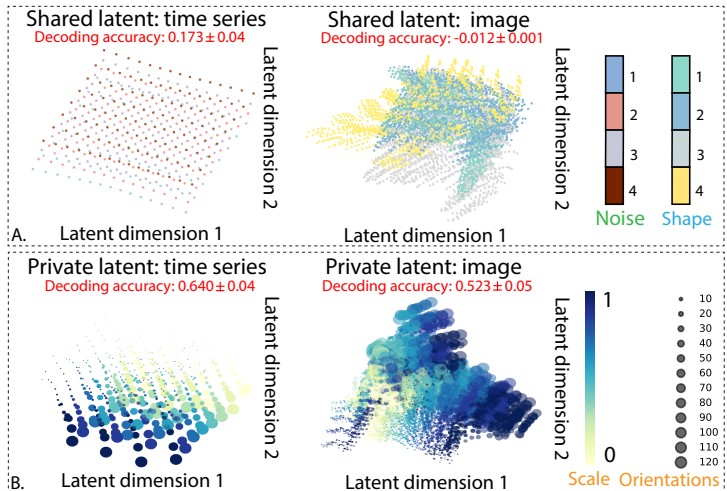

Figure 14: PCA embeddings for shared and private behavioral and neural latent subspace: A. Shared latent spaces for each modality, each point represents a single frame, color-coded by the corresponding private features: the private latent spaces retains the private features. B. Private latent for each modality, color-coded by orientations and scales.

Table 4: Shared-AE unpaired vs paired results: 1. The shared latent results in better classification accuracy in shared features while private latent retained more modality-specific information. 2. Shuffling one of the modalities during test time does not affect latent classification accuracy which indicates that there is no information leakage between modalities.

|  | Shared-AE (unpaired) | Shared-AE (paired) |
|---|---|---|
| Private time series latents on scale and orientations↓ | $0.647 \pm 0.05$ | $0.640 \pm 0.04$ |
| Shared time series latents on scale and orientations↑ | $0.980 \pm 0.09$ | $0.981 \pm 0.01$ |
| Private time series latents on shape↓ | $0.017 \pm 0.001$ | $0.008 \pm 0.001$ |
| Shared time series latents on shape↓ | $-0.012 \pm 0.001$ | $-0.015 \pm 0.01$ |
| Private time series latents on temporally-structured noise↑ | $0.732 \pm 0.012$ | $0.745 \pm 0.005$ |
| Shared time series latents on temporally-structured noise↓ | $0.183 \pm 0.01$ | $0.173 \pm 0.04$ |
|  |  |  |
| Private image latents on scale and orientations↓ | $0.523 \pm 0.02$ | $0.53 \pm 0.05$ |
| Shared image latents on scale and orientations↑ | $0.757 \pm 0.08$ | $0.751 \pm 0.07$ |
| Private image latents on shape↑ | $0.768 \pm 0.03$ | $0.767 \pm 0.09$ |
| Shared image latents on shape↓ | $0.288 \pm 0.01$ | $0.289 \pm 0.008$ |
| Private image latents on temporally-structured noise↓ | $0.012 \pm 0.05$ | $0.011 \pm 0.08$ |
| Shared image latents on temporally-structured noise↓ | $0.015 \pm 0.01$ | $0.015 \pm 0.007$ |

Table 5: Baseline model performance on unpaired and paired datasets: the drop on classification indicates that there is information leakage during training

| **Baseline comparison for unpaired dataset** | MM-VAE | Joint encoding model |
|---|---|---|
| Shared time series latents on scale and orientations | $0.52 \pm 0.21$ | $0.58 \pm 0.1$ |
| Shared time series latents on shape | $-0.01 \pm 0.04$ | $0.013 \pm 0.04$ |
| Shared time series latents on temporally-structured noise | $-0.001 \pm 0.007$ | $0.034 \pm 0.08$ |
| Shared image latents on scale and orientations | $0.42 \pm 0.11$ | $0.55 \pm 0.11$ |
| Shared image latents on shape | $0.146 \pm 0.001$ | $0.010 \pm 0.01$ |
| Shared image latents on temporally-structured noise | $-0.003 \pm 0.007$ | $0.042 \pm 0.07$ |
|  |  |  |
| **Baseline comparison for paired dataset** | MM-VAE | Joint encoding model |
| Shared time series latents on scale and orientations | $0.90 \pm 0.06$ | $0.975 \pm 0.01$ |
| Shared time series latents on shape | $0.22 \pm 0.03$ | $-0.012 \pm 0.01$ |
| Shared time series latents on temporally-structured noise | $0.0005 \pm 0.000$ | $0.024 \pm 0.001$ |
| Shared image latents on scale and orientations | $0.90 \pm 0.06$ | $0.646 \pm 0.001$ |
| Shared image latents on shape | $0.154 \pm 0.003$ | $0.861 \pm 0.003$ |
| Shared image latents on temporally-structured noise | $0.003 \pm 0.000$ | $0.071 \pm 0.003$ |

Table 6: Decoding accuracy for temporally-structured noise (private time-series feature) on simulated dataset

| Subspace | PSID | DPAD | **Shared-AE** |
|---|---|---|---|
| Shared latent↓ | $0.99 \pm 0.00$ | $0.99 \pm 0.00$ | $0.015 \pm 0.01$ |
| Private image latent↓ | NA | NA | $0.012 \pm 0.05$ |
| Private time-series latent↑ | NA | NA | $0.732 \pm 0.00$ |

## A.13 ADDITIONAL RESULTS: **headfixed dataset**

### A.13.1 BASELINE MODEL CONFIGURATION

In MM-GP-VAE, the dimensions of both the shared and private latent spaces are set to 50, similar to configurations in MM-VAE and the Joint Encoding model. For PSID: $n1 = 5, nx = 50, i = 2$. For DPAD: $n1 = 5, nx = 50$ and the method code is $'NDM_C zNonLin'$. These consistent latent space dimensions across models provide a standardized basis for comparison, ensuring that the

Table 7: Baseline model without CS-divergence vs. Shared-AE: performance on simulated paired datasets

| Decoding latents and targets | Baseline | Shared-AE |
|---|---|---|
| Shared time series latents on scale and orientations↑ | $-0.036 \pm 0.01$ | $0.981 \pm 0.01$ |
| Shared time series latents on shape ↓ | $-0.012 \pm 0.01$ | $-0.015 \pm 0.01$ |
| Shared time series latents on temporally-structured noise ↓ | $-0.018 \pm 0.005$ | $0.173 \pm 0.004$ |
| Shared image latents on scale and orientations↑ | $0.67 \pm 0.04$ | $0.751 \pm 0.07$ |
| Shared image latents on shape ↓ | $0.52 \pm 0.005$ | $0.289 \pm 0.008$ |
| Shared image latents on temporally-structured noise↓ | $0.035 \pm 0.003$ | $0.015 \pm 0.007$ |
| Private time series latents on scale and orientations↓ | $-0.030 \pm 0.01$ | $0.674 \pm 0.05$ |
| Private time series latents on shape ↓ | $-0.023 \pm 0.01$ | $0.017 \pm 0.001$ |
| Private time series latents on temporally-structured noise ↑ | $-0.004 \pm 0.001$ | $0.732 \pm 0.0012$ |
| Private image latents on scale and orientations↓ | $0.70 \pm 0.02$ | $0.523 \pm 0.02$ |
| Private image latents on shape ↑ | $0.78 \pm 0.005$ | $0.768 \pm 0.03$ |
| Private image latents on temporally-structured noise↓ | $0.075 \pm 0.006$ | $0.012 \pm 0.05$ |

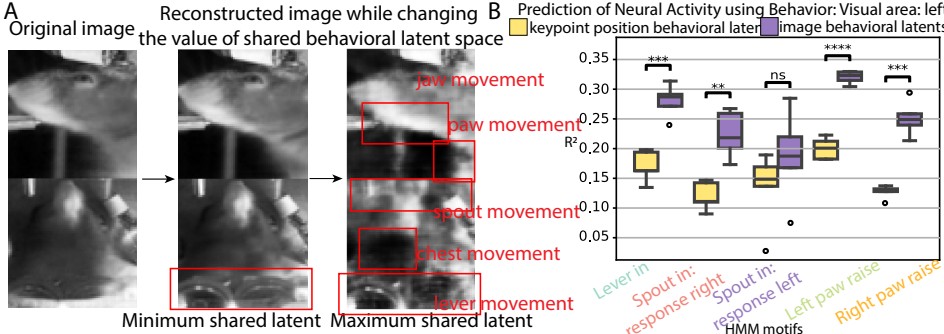

Figure 15: **Head-fixed mouse: behavioral video and WFCI**: A. Latent traversal for the shared behavioral latents shows automatic identification of neurally-relevant features such as jaw and paw movement. B. Neural activity prediction: a comparison between the video-based behavioral latents and the keypoint behavioral latents shows that there is more shared information about behavioral videos than poses in select brain regions. Full list shown in Appendix A.11

evaluation focuses on differences in model design and methodology rather than variations in latent space capacity.

### A.13.2 BASELINE COMPARISONS

Five baseline models are applied to the 2AFC simultaneously collected neural WFCI and behavioral video data for comparison: PSID Sani et al. (2021), DPAD Sani et al. (2024), MM-VAE Shi et al. (2019), the Joint Encoding Model Singh Alvarado et al. (2021), and MM-GP-VAE Gondur et al. (2024). Importantly, the first four baseline models failed to separate the latent space into distinct shared and private latent subspaces. In order to quantify the amount of information represented in each latent subspace, we compared the ability of each model to reconstruct one of the modalities, the body positions, in Tables 11 and 12 Shared-AE achieves higher decoding accuracy, particularly on unpaired datasets, compared to existing models. This demonstrates that it does not suffer from modality leakage, ensuring that during inference, robust representations can be generated even if only one modality's data is available. This is particularly advantageous in situations where acquiring all modalities simultaneously is challenging. The ability to produce meaningful latents without the need for all modalities underscores the practical utility and flexibility of our approach. We emphasize that Shared-AE offers significantly improved interpretability of the latent space. Moreover, unlike PSID, Shared-AE is capable of handling image data and can accommodate more than three modalities, providing greater flexibility for complex multimodal tasks.

The model proposed by Gondur et al. (2024) combines the behavioral and the neural latent into a common subspace. This can be detrimental to interpretability since it is unclear which modality is leading to the performance of the shared latent. As detailed in Appendix A.3, the combined shared

latent causes information bleeding between different modalities. We implemented MM-GP-VAE based on Gondur et al. (2024) as well as a similar model by incorporating a single shared latent space for both modalities and tested it on the 2AFC dataset, referring to this implementation as MM-nonGP-VAE. These models were used as baselines to compare against our proposed approach. As shown in Table 8, Shared-AE outperforms both of these models on both paired and unpaired tasks. Critically, Shared-AE has the ability to generate reasonable latents despite corrupted data in the other modality during inference.

Table 8: Behavioral decoding accuracy with unpaired modalities for 2AFC body position dataset

| tasks | MM-GP-VAE | MM-nonGP-VAE | Shared-AE |
|---|---|---|---|
| Private latent | $-0.01 \pm 0.00$ | $0.25 \pm 0.02$ | $0.22 \pm 0.03$ |
| Shared latent | $0.006 \pm 0.00$ | $0.23 \pm 0.03$ | $0.41 \pm 0.05$ |

Table 9: Behavioral decoding accuracy with paired modalities for 2AFC body position dataset

| tasks | MM-GP-VAE | MM-nonGP-VAE | Shared-AE |
|---|---|---|---|
| Private latent | $0.37 \pm 0.00$ | $0.24 \pm 0.02$ | $0.22 \pm 0.03$ |
| Shared latent | $0.36 \pm 0.01$ | $0.32 \pm 0.03$ | $0.41 \pm 0.05$ |

For comparison with PSID, we further validate that PSID and DPAD cannot effectively handle image data, we conducted an experiment where the flattened behavioral image was used as the input behavioral data for PSID and DPAD (Table 10). We then used the generated neural latent representations for body position decoding and compared the results with those obtained from Shared-AE, which directly utilizes the behavioral image as input. The decoding results obtained by Shared-AE outperformed the rest.

Table 10: Behavioral decoding accuracy with paired modalities for 2AFC image dataset

| Subspace | PSID | DPAD | Shared-AE |
|---|---|---|---|
| Shared latents | $0.32 \pm 0.02$ | $0.31 \pm 0.00$ | $0.38 \pm 0.01$ |

Table 11: Behavioral decoding accuracy with unpaired modalities for 2AFC body position dataset

| Subspace | MM-VAE | Joint Encoding Model | MM-GP-VAE | Shared-AE |
|---|---|---|---|---|
| Private latents | $NA$ | $NA$ | $-0.01 \pm 0.00$ | $0.22 \pm 0.03$ |
| Shared latents | $0.22 \pm 0.03$ | $0.20 \pm 0.02$ | $0.006 \pm 0.00$ | $0.41 \pm 0.05$ |

Table 12: Behavioral decoding accuracy with paired modalities for 2AFC body position dataset

| Subspace | PSID | DPAD | MM-VAE | Joint Encoding Model | MM-GP-VAE | Shared-AE |
|---|---|---|---|---|---|---|
| Private latents | $NA$ | $NA$ | $NA$ | $NA$ | $0.37 \pm 0.00$ | $0.22 \pm 0.03$ |
| Shared latents | $0.20 \pm 0.03$ | $0.27 \pm 0.02$ | $0.41 \pm 0.07$ | $0.40 \pm 0.06$ | $0.36 \pm 0.01$ | $0.41 \pm 0.05$ |

### A.13.3 UMAP EMBEDDINGS OF THE PRIVATE LATENT SUBSPACE

We show that the private latent subspaces cannot capture different HMM states inferred by behavioral latents (Fig. 16).

### A.13.4 INFLUENCE OF THE PRIVATE BEHAVIORAL LATENT SUBSPACE ON IMAGE RECONSTRUCTION

We performed the same task as in Sec. 4.2, varying the private behavioral latent from its minimum value to its maximum value. (Fig. 17).

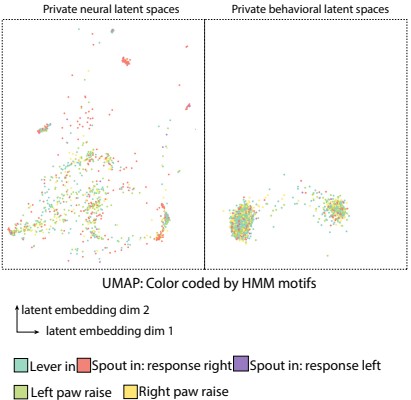

Figure 16: UMAP embeddings for private behavioral and neural latent subspace for the 2AFC dataset.

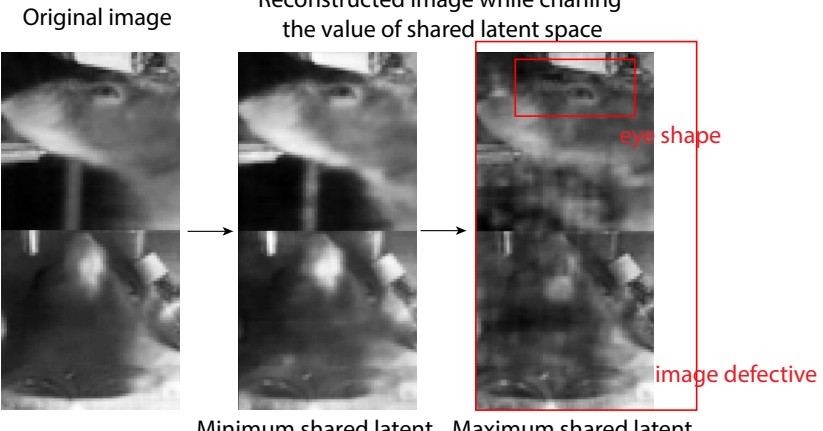

Figure 17: Changing the value of the private latent space from its minimum to its maximum results in a defection in the image. The private latent captures more about the behaviorally unrelated features.

