# OpenReview forum: "Shared-AE: Automatic Identification of Shared Subspaces in High-dimensional Neural and Behavioral Activity"
_ICLR.cc/2025/Conference — ICLR 2025 Poster_

### Official Review · Reviewer_HB99 · 2024-10-18

**Soundness:** 3
**Presentation:** 3
**Contribution:** 3
**Rating:** 8
**Confidence:** 4

**Summary:**

This paper introduces a latent-space disentangling autoencoder to identify shared and private latent features from multi-modal neural and behavioral data. Proposed disentanglement is based on a Cauchy-Schwarz divergence based regularizer applied between different components (private and shared features) of the latent representations obtained via behavioral and neural encoders. Both inter and intra modality regularization losses are combined in addition to the standard autoencoder loss. Experimental analyses are performed first on a simulated dataset, and then on different complex behavioral datasets with neural recordings from mice.

**Strengths:**

- Consistent experimental findings based on a simple disentangled representation learning model with a tailored training objective.
- It has a unique empirical strength with a focus on neural data analysis from complex multi-modal social behavior experiments.

**Weaknesses:**

- Some of the empirical results need further validation, considering the details present in the Appendix.

**Questions:**

- The overall idea of disentangling the latent representation space using inter- and intra-modality loss regularizers has been previously explored in several works. There are also actually works proposing a similar autoencoder regularization framework in other settings [Tran et al. "Cauchy–Schwarz Regularized Autoencoder", JMLR 2022]. Perhaps one question that the authors should clarify with a clear statement is their methodological ML novelty (i.e., if the proposed regularized training scheme is completely novel, or if the paper only contains a strong empirical novelty).

- Majority of the results show strong consistency between the disentangled latent features extracted from behavioral and neural data. Regarding the latent space visualizations (UMAP etc), how did the authors determine the latent dimensionality in each experiment? How consistent are these results with respect to changing this dimensionality?

- The dataset retrieved for the 2AFC experiments seems rather small in terms of the number of trials. Also it seems to be divided only once into a train/test split. Therefore, I would ask if the authors performed any CV of the model training process, and evaluate the significance of their results in that sense?

---

> ### Author Response · Authors · 2024-11-23
>
> We appreciate your recognition of the unique empirical strength of our model, particularly in the context of analyzing neural data from complex, multi-modal social behavior experiments.
>
> Weakness 1:
>
> The application of CS-divergence in Tran et al. regularizes the latent distribution using a **pre-defined prior distribution**. However, in this case, we are trying to minimize the distance between **two learned distributions**. This is a critical departure from previous studies, and a technical novelty of this paper.
>
> Additionally, we do not just show the utility of the CS divergence between two learned distributions, but also present a way to separate the latent features into modality-specific yet shared subspaces.
>
> Traditional multimodal approaches lack an explicit orthogonal constraint to ensure that each latent subspace captures distinct information. For instance, the CLIP model uses a contrastive loss between two latent spaces but does not account for private features, which may result in overlapping information. To address this, we introduce an inverse loss on the latent subspaces to enforce orthogonality between the shared and private components. This not only enhances the training process but also provides richer insights into the underlying data.
>
> Finally, the paper also pioneers the ability to extract shared features between more than two modalities.
>
> Weakness 2:
>
> We choose the latent dimension based on the reconstruction results: the smallest value when the MSE loss converged. We evaluated the decoding accuracy of body positions using shared neural latents with varying latent dimensions: 50, 80, and 100. As shown in Table 8, the decoding accuracy remains relatively consistent across these dimensions, indicating that the model's performance is robust to changes in latent dimensionality.
>
> Weakness 3:
>
> We apologize for the lack of clarity on this topic. We evaluated the latent dimension based on reconstruction accuracy using 5-fold cross-validation. The held-out data served as the test set on which all the results are reported.

---

> > ### Comment · Reviewer_HB99 · 2024-11-25
> > **Response to Authors' Rebuttal**
> >
> > Thanks to the authors for their efforts during the rebuttal to refine their work and clarity.
> >
> > The difference in the training/regularization strategy to existing methods is now more clear to me, thanks for the clarification. This supports the current submission's methodological contribution as well, which was previously not very clear as far as I saw from the other reviewers' comments too.
> >
> > In the light of the rebuttal clarifications, I think this work contains both a technical and an empirical novelty in a well written manuscript. Thus, I will keep my score for an accept.

---

### Official Review · Reviewer_h2Dv · 2024-10-23

**Soundness:** 3
**Presentation:** 2
**Contribution:** 2
**Rating:** 6
**Confidence:** 4

**Summary:**

This paper contributes to a growing literature on learning shared representations for multimodal data in neuroscience, where many researchers are interested in learning joint representations of brain data and behavior. Whereas many previous methods have focused on learning shared latent representations by combining latents learned from individual modalities, this work further partitions each modality's latent space into a private latent spaces and a shared latent space, which is linearly mixed with other modalities' shared latents. This separation is engendered by the use of a Cauchy-Schwarz Divergence for aligning shared latents and separating shared from private latents. Experiments on one synthetic and two real data sets show suggestive links between brain data and behavior, though what one is to make of these is a bit unclear. Moreover, the technical contribution of the paper is perhaps small when considered in light of other work cited.

In all, this is a solid paper that is, in my view, below the bar for acceptance without a more substantial technical contribution.

**Strengths:**

- Principled approach to structuring latent spaces based on a desired semantics: some information in each latent space is common to all modes, some is private.
- The need for interpretable joint encodings is of high interest in neuroscience.
- CS-Divergence is a reasonable means of effecting the separation of subspaces, and the authors have chosen a pretty reasonable-seeming method of approximating this quantity.
- The experiments on real data use challenging datasets that encapsulate many challenges faced by the community.

**Weaknesses:**

- It is somewhat unclear what the technical innovation in this paper is beyond the Yi et al. preprint cited, as well as a similar paper by Yi and Saxena at EMBC in 2022 [1]. Both of those works use the same CS divergence setup as here, and I am struggling to see where the technical innovation is (though the application is somewhat different). I don't see the strength of the experimental results here being novel or interesting enough on their own to justify acceptance without an additional technical advance.
- The authors use a latent space partition that is distinct from the Whiteway et al. paper but somewhat related to the Sani et al. work they cite. I realize that the PSID paper is linear, but the Shanechi group also has work on nonlinear methods that preserve this kind of partition (the most recent of which was likely unpublished when this work was submitted) that should probably come in for a fuller discussion.
- In the framing of this work, I don't believe I fully understood the authors' rationale for needing shared vs. private subspaces. It's conceptually interesting, but the experiments simply focus on decoding. In what circumstances do we need such a partition, and what is the sign that not having it is hurting us scientifically? If this were clearer, I think it would be easier for readers to judge the success of the experiments.
- The figures are all quite small and cramped, making them somewhat hard to parse. It's not always clear what the "win" is with the experiments.

[1]: D. Yi and S. Saxena, "Modeling the behavior of multiple subjects using a Cauchy-Schwarz regularized Partitioned Subspace Variational AutoEncoder (CS-PS-VAE)," 2022 44th Annual International Conference of the IEEE Engineering in Medicine & Biology Society (EMBC), Glasgow, Scotland, United Kingdom, 2022, pp. 497-503, doi: 10.1109/EMBC48229.2022.9871466.

**Questions:**

## My key question:
- What is the technical innovation between this and the two other Yi papers? Is it just a different application?

## Small points of clarification:
- ll. 108-120: the ending here is a bit vague; would help to clarify what these sorts of models would miss in more complex tasks (and hopefully show in experiments)
- ll. 141-152: Model scales as number of pairs of modalities; probably not a limitation in practice, but a few words about scaling might help.
_ l. 165: what is $s'_t$ here? Is it the same as $s_t^{pre}$?
- l. 171: what is $y'$? Is the prime a typo?
- ll. 162-168: It would be nice to have a diagram of this, since one could easily lose track of the different linear models: If I understand correctly: $s^{pre}$ is a linear function of each modality's latents, and $s^i$ is a linear function of both modalities' _shared_ latents.
- l. 196 In Eq 2, how well does this estimator do in moderate-sized latent spaces? Is it a reasonable estimator? One is effectively saying that the density belongs to a reproducing kernel Hilbert space, right?
- ll. 259-264: This description is a bit terse and may be hard to follow for readers (like me) who were not familiar with this dataset. I realize details are in the supplement, but the main text should be a bit more self-contained.
- ll. 318-321: Why, exactly, do we need a strong separation between modalities? What is the use case? I realize it affects decoding performance (e.g., Figure 3) but what might we use this analysis to conclude in an experiment?
- Figure 4A: Sorting the rows and columns by some sort of biclustering algorithm might make the correlation structure more apparent. This matrix plot is not very compelling as presented.

---

> ### Author Response · Authors · 2024-11-23
>
> We thank the reviewer for their detailed comments and feedback. We appreciate your recognition of our proposed autoencoder approach for uncovering shared and private latent spaces in neural and behavioral data. The use of Cauchy-Schwarz (CS) divergence to regularize the shared and private subspaces is a key aspect of our model, allowing us to effectively identify features common across modalities as well as those unique to each modality.
>
> Weakness 1:
>
> Yi et al. and Yi and Saxena do indeed use a CS-divergence, but in a drastically different way than in our study: they use CS-divergence to fit the latent distribution to a **predefined prior distribution**, whereas Shared-AE minimizes the distance between two **learned** distributions instead. This approach avoids the limitations of predefined priors and allows for more flexible and meaningful representations. This is a critical departure from previous studies, and a technical novelty of this paper. Moreover, we do not just show the utility of the CS divergence between two learned distributions, but also present a way to separate the latent features into modality-specific yet shared subspaces. Additionally, the paper also pioneers the ability to extract shared features between more than two modalities.
>
> Weakness 2:
>
> As shown in Table 4, compared to DPAD by the Shanechi group, our model produces high decoding accuracy for the 'unpaired' task setting (see global rebuttal). We emphasize that our model offers significantly improved interpretability of the latent space. Moreover, unlike PSID, our model is capable of handling image data (Table 5) and can accommodate more than three modalities, providing greater flexibility for complex multimodal tasks.
>
> The key advantage of the Shared-AE model over other approaches lies in its ability to effectively disentangle shared and private features, resulting in a clearer separation of modality-specific information. This distinct separation not only enhances the interpretability of the latent representations but also supports a wider range of downstream tasks across different data modalities.
>
> Weakness 3:
>
> The purpose of introducing a private latent space is to preserve unique features for each modality, allowing for a more interpretable shared latent space that focuses solely on the common information between the modalities. For instance, if there is extraneous information in behavioral recordings that is not being encoded by certain regions of the brain, this information will still be able to be retained in the private latent subspace, which is helpful for a full understanding of behavior. Similarly, if there are brain regions that are not participating in a certain behavior, their activity may be lost in alternative approaches without the private subspaces. By isolating private features, this approach enables a clearer understanding of the distinct contributions of various brain areas during different behavioral states.
>
> Weakness 4:
>
> Thank you for this insight; we will make each figure larger and less cramped when we update the manuscript. We have listed in our original manuscript a few scientific takeaways from Shared-AE applied to 2AFC and social behavior data. Importantly, the neural modality that we consider in these experiments is WFCI neural data, which records large parts of the entire dorsal cortex of the mouse. There is a large diversity of regions in this modality, and these data have historically been collected primarily during a head-fixed task. Here, while we first analyze WFCI in the more stereotyped 2AFC head-fixed task setting and recapitulate key results from previous studies that have also examined these settings, the novel experiment and analysis is of WFCI during freely-moving, unstructured social behavior between two mice. This recording condition is very under-explored, and provides a case where we do not have many preconceived ideas about how the different regions may be encoding not just the mouse's own behavior but that of the other mouse. Shared-AE gives us a very clear and compact representation of how much each mouse's behavioral features are represented in the recorded neural activity, and vice versa - how much each brain region may encode the social behavior. These insights can be validated using optogenetic silencing or behavioral manipulations in future studies.

---

> > ### Comment · Reviewer_h2Dv · 2024-11-25
> >
> > I appreciate the authors' clarifications, as well as the additional decoding experiments comparing to other models. I will raise my score once I have had a chance to discuss with other reviewers in the next phase of the process.

---

> ### Author Response · Authors · 2024-11-23
>
> Small points:
> Thank you for bringing those points to our attention. We have addressed them thoroughly in the main text and made the necessary revisions to ensure clarity and completeness.
>
> Point 7: Why, exactly, do we need a strong separation between modalities? What is the use case? I realize it affects decoding performance (e.g., Figure 3) but what might we use this analysis to conclude in an experiment?
>
> The key advantage of the Shared-AE model over other approaches lies in its ability to effectively disentangle shared and private features, resulting in a clearer separation of modality-specific information. This distinct separation not only enhances the interpretability of the latent representations but also supports a wider range of downstream tasks across different data modalities. As shown in Table 6, when the shared features across both modalities are weak, both PSID and DPAD struggle to effectively separate the private features from the shared latent spaces. Here, both models resulted in an erroneously high decoding accuracy of a private feature from the shared latent space. Shared-AE was able to successfully decouple the shared vs. private features in its latent subspaces.

---

### Official Review · Reviewer_QBbN · 2024-11-01

**Soundness:** 2
**Presentation:** 3
**Contribution:** 2
**Rating:** 3
**Confidence:** 4

**Summary:**

The paper proposes an autoencoder-based framework for finding shared and private subspaces of multimodal neural and behavioral data. Their goal is to separate the shared subspace from the modality-specific subspaces. To do so, constraints (Cauchy-Schwarz divergence) are added on the distribution of subspaces to encourage/discourage their alignment for this particular goal. This method is evaluated on one simulated dataset as well as two distinct experimental datasets from mice.

**Strengths:**

1.	The paper is well-presented. The goal is clear and many analyses including both simulated datasets and two experimental datasets with different behavioral complexities are analyzed.
2.	Various analyses such as connecting behavioral variates to corresponding brain area through shared latents, and investigating the difference between modeling behavior as markers or raw videos in the learned shared features.

**Weaknesses:**

1.	Methodological novelty seems minimal. The authors note “a novel regularization term designed to identify features common to both behavior and neural activity” as their main methodological novelty. However, a very similar regularization scheme has been previously proposed by Yi et al. (2022). The difference between Shared-AE and this work is not adequately discussed making the methodological novelty unclear. Also, the idea of using CS-divergence instead of standard VAE with KL-divergence is not novel either as previously proposed by Tran et al. (2021).

2.	There are numerous methods on neural-behavioral modeling and finding shared vs. private subspaces, none of which are compared to and many which are not discussed. In general, the manuscript seems to mix up unsupervised latent variable models of neural data with latent variable models of neural-behavioral data in its discussions and writeup. The only neural-behavioral data discussed in Related work (but not compared to) is Schneider et al 2023. Another neural-behavioral model that is cited is Sani et al 2021, but it is not discussed or compared to, and is instead grouped with an unsupervised model of neural data. Another neural-behavioral model in Zhou and Wei 2020 is also simply cited but not compared to. The authors need to separate the neural vs. neural-behavioral models in their manuscript and provide sufficient discussion of differences between other neural-behavioral models with theirs. Comparison to these neural-behavioral models is also needed to show the advantage of this method. In addition to the above cited works, there are also some other very relevant neural-behavioral models that are not cited, for example:

Gondur et al. 2024: This work appeared in the previous ICLR and has a very similar architecture designed for the same purpose using GP-VAE. However, it is not cited, and the key differences are not discussed. Given how closely related this method is to the authors’ work, it can serve as a baseline.

Hurwitz et al. 2021: This work proposes a sequential VAE for modeling neural-behavioral data. This needs to be cited and discussed.

Sani et al. 2024: This work proposes an RNN-based architecture that separates shared/private subspaces in neural-behavioral data and needs to be cited and discussed.

3.	Effect of novel terms in the loss i.e., the CS-divergence and their inverses are not assessed. As this is the main addition to a standard multimodal AE architecture in this work, it is crucial to evaluate whether presence of each term contributes to current results or not. Even without these constraints, the reconstruction loss itself can enforce shared vs private subspaces (at least to some extent) as the shared ones reconstruct both modalities whereas private ones reconstruct the specific modality alone.


4.	Lack of baseline comparison in real data analysis. The same baselines used in simulated dataset (Shi et al (2019), Singh Alvarado et al. (2021)) are not shown in real data. Additionally, there are several relevant works on neural-behavioral modeling some of which could be used as baseline to better assess what benefits Shared-AE adds as mentioned in item 2 above.

5.	The authors claim that their framework is better for more complex/social behavior types than Schneider et al. (2023). But what about all the other neural-behavioral models? Is shared-AE more suitable for complex behavior than others and if so why? This claim does not seem convincing without further comparisons.

6.	I find calling this method unsupervised very misleading. In the context of neural-behavioral modeling, supervision typically means use of behavior for guiding behavior-related features of neural activity. In this sense, the proposed approach is fully supervised not only during learning but also during inference, putting it in the multimodal family. The manuscript refers to this method as unsupervised throughout the paper including in the title. This needs to be corrected.

7.	The model uses hyperparameters $(\alpha, \beta, \gamma, \delta)$ to control the contribution of regularizations to the overall loss. However, the effect of these hyperparameters on the results are not investigated. Authors note that the results are robust to changes in the hyperparameters, but I did not find the results that show this robustness. Please provide these.


Minor weaknesses/questions

1.	Why does the method need to learn two separate shared latents? It seems these two should ideally correspond to the same thing. Why not have a single shared latent which is used in both decoders?

2.	In the unpaired analysis, is shuffling happening across time? Why does maintaining performance in this scenario indicate avoiding modality leakage?

3.	What is the basis for choosing the state dimension based on Fig. 8? Why are the reconstruction performance vs dimension so noisy in Fig. 8?

4.	What does min/max R2 refer to in Fig. 8?

typographical errors:

-	Line 302: Fig. 4.1 => Fig. 4?
-	Line 302: missing space between “data” and “(Fig”
-	Fig. 7 caption includes panels E-F while the results are missing.  “E-F: Prediction accuracy for neural activity and behavior under different distance groups.” It seems these panels are not included.
-	Captions for panels B and C of Fig. 10 do not match. It seems the order is wrong.
-	Fig 11 has very tiny titles
-	Line 1002: reference to Fig. A.9.3 is incorrect.

References:

Daiyao Yi, Simon Musall, Anne Churchland, Nancy Padilla-Coreano, and Shreya Saxena. Disentangled multi-subject and social behavioral representations through a constrained subspace variational autoencoder (cs-vae). bioRxiv, 2022. doi: 10.1101/2022.09.01.506091. URL https: //www.biorxiv.org/content/early/2022/09/05/2022.09.01.506091.

Linh Tran, Maja Pantic, and Marc Peter Deisenroth. Cauchy-schwarz regularized autoencoder, 2021. URL https://arxiv.org/abs/2101.02149.

Yuge Shi, N. Siddharth, Brooks Paige, and Philip H. S. Torr. Variational mixture-of-experts autoencoder for multi-modal deep generative models, 2019. URL https://arxiv.org/abs/1911.03393.

Jonnathan Singh Alvarado, Jack Goffinet, Valerie Michael, William Liberti, Jordan Hatfield, Timothy Gardner, John Pearson, and Richard Mooney. Neural dynamics underlying birdsong practice and performance. Nature, 599(7886):635—639, November 2021. ISSN 0028-0836.

Steffen Schneider, Jin Hwa Lee, and Mackenzie Weygandt Mathis. Learnable latent embeddings for joint behavioural and neural analysis. Nature, 617: 360–368 May 2023. ISSN 1476-4687.

Rabia Gondur, Usama Bin Sikandar, Evan Schaffer, Mikio Christian Aoi, and Stephen L Keeley. Multi-modal gaussian process variational autoencoders for neural and behavioral data. In International Conference on Learning Representations, 2024.

Cole Hurwitz, Akash Srivastava, Kai Xu, Justin Jude, Matthew Perich, Lee Miller, and Matthias Hennig. Targeted neural dynamical modeling. Advances in Neural Information Processing Systems, 34:29379–29392, 2021.

Omid G Sani, Hamidreza Abbaspourazad, Yan T Wong, Bijan Pesaran, and Maryam M Shanechi. Modeling behaviorally relevant neural dynamics enabled by preferential subspace identification. Nature Neuroscience, 24(1):140–149, 2021.

Ding Zhou and Xue-Xin Wei. Learning identifiable and interpretable latent models of high-dimensional neural activity using pi-vae.  Advances in Neural Information Processing Systems, volume 33, pp. 7234–7247,  2020.

Omid G Sani, Bijan Pesaran, and Maryam M Shanechi. Dissociative and prioritized modeling of behaviorally relevant neural dynamics using recurrent neural networks. Nature Neuroscience, 27: 2033–2045, 2024.

**Questions:**

My questions are the ones raised in weaknesses and Minor weaknesses/questions.

---

> ### Author Response · Authors · 2024-11-23
>
> We thank the reviewer for their detailed comments and feedback. We appreciate your recognition of the strengths of our work and the constructive suggestions for improvement. Below, we address each of your concerns:
>
> Weakness 1:
>
> The application of CS-divergence in Yi et al. and also in Tran et al. regularizes the latent distribution using a **pre-defined prior distribution**. However, in this case, we are trying to minimize the distance between **two learned distributions**.
>
> This is a critical departure from previous studies and a technical novelty of this paper. Moreover, we do not just show the utility of the CS-divergence between two learned distributions, but also present a way to separate the latent features into modality-specific yet shared subspaces. Additionally, the paper pioneers the ability to extract shared features between more than two modalities.
>
> Weakness 2:
>
> We acknowledge that our discussion in the related work section did not clearly distinguish between unsupervised latent variable models of neural data and latent variable models of neural-behavioral data. We will revise this section to explicitly separate these categories and provide a more structured comparison.
>
> We appreciate your feedback regarding the need for comparisons with other neural-behavioral models. We conducted comprehensive evaluations on the 2AFC dataset, demonstrating that Shared-AE achieves higher decoding accuracy, particularly on unpaired datasets, compared to existing models. To evaluate decoding accuracy, we trained a linear regression model on the training dataset to predict body position using the neural latent representations. In cases where a decoder was readily available during training, such as in PSID and DPAD, we directly used it to generate these predictions.
>
> In comparison with other models, we emphasize that Shared-AE has better decoding accuracy and offers significantly improved interpretability of the latent space. Moreover, unlike PSID, Shared-AE can handle image data and accommodate more than three modalities, providing greater flexibility for complex multimodal tasks. To further validate that PSID cannot effectively handle image data, we conducted an experiment where the flattened behavioral image was used as the input behavioral data for PSID and DPAD (Table 5). We then used the generated neural latent representations for body position decoding and compared the results with those obtained from Shared-AE, which directly utilizes the behavioral image as input. The decoding results obtained by Shared-AE outperformed the rest.
>
> The key advantage of the Shared-AE model over other approaches lies in its ability to effectively disentangle shared and private features, resulting in a clearer separation of modality-specific information. This distinct separation not only enhances the interpretability of the latent representations but also supports a wider range of downstream tasks across different data modalities. As shown in Table 6, when the shared features across both modalities are weak, both PSID and DPAD struggle to effectively separate the private features from the shared latent spaces. Here, both models resulted in an erroneously high decoding accuracy of a private feature from the shared latent space. Shared-AE was able to successfully decouple the shared vs. private features in its latent subspaces.
>
> Weakness 3:
>
> As a baseline, we trained the model on a simulated dataset using the same architecture but without applying CS-divergence. As shown in Table 7, the results indicate a lack of separation between the private and shared latent spaces, highlighting the necessity of constraints to effectively distinguish shared and private features. Importantly, the shared subspaces in the Shared-AE are not in fact reconstructing both modalities; they remain completely separate from each other, with purely the CS-divergence to regularize them to be similar.
>
> Weakness 4:
>
> Thank you for the suggestion; we have now added these relevant baselines, see our response to Weakness 2.
>
> Weakness 5:
>
> Shared-AE can handle image data for behavioral modeling as well as more than two modalities. These are key capabilities that have not been shown directly in other models. As shown in Fig. 6B, using the behavioral images leads to better prediction accuracy on neural activity. This indicates that the image is better than only the pose estimation for representing the behavior, which is consistent with the findings reported in the original paper (Musall et al., 2019). Additionally, Shared-AE can quantify the relationship between the brain and complex behaviors, whereas PSID lacks interpretability on the generated latent space since it does not separate the shared latents from each modality, thus making it unclear which modality the shared latents are arising from.

---

> ### Author Response · Authors · 2024-11-23
>
> Weakness 6:
>
> Thank you for the input. Shared-AE can indeed generate latent representations during inference with only one modality. However, we do agree that calling it 'unsupervised' is a bit misleading, and we will change this terminology.
>
> Weakness 7:
>
> As shown in Appendices 8.2 and 8.3, the results remain robust as long as the CS loss decreases below a certain threshold.
>
> Minor:
> 1. Thank you for allowing us to clarify this point. Shared-AE explicitly learns modality-specific shared latent subspaces. This separation preserves interpretability by reducing information leakage across modalities, allowing us to better disentangle distinct modality-specific features. Indeed, having common shared subspaces can be detrimental towards interpretability, since it is unclear which modality is leading to the performance of the shared latent. As an additional key consequence of modality-specific shared subspaces, if we only have data from one modality during inference, we do not require data from the other modalities to generate robust and meaningful representations. This is an important point, so please let us know if we can clarify it further.
>
> 2. Here, as also included in the global rebuttal, we designed an 'unpaired' task to ensure that there is no information leakage between different modalities. Critically, after training the model, if one modality's data is corrupted during inference, this should not affect the other modality's latent representation or decodability. Here, during inference only, one modality was shuffled across time, while the other modality remained unchanged. The resulting `unpaired' data was then input into the trained model to generate new latent representations for the unshuffled modality. This approach allowed us to assess whether the model could maintain the integrity of the unshuffled modality's latent features, ensuring that the latent representations were unaffected by the shuffled modality.
>
> 3. We evaluated the latent dimension based on reconstruction accuracy using 5-fold cross-validation. We chose the minimum latent dimension when both the neural and behavioral reconstruction accuracy reached a plateau. The noisy performance is due to the trial-to-trial variability.
>
> 4. In Fig. 8, the mean R² represents the average accuracy across all keypoint positions and neural channels, while the max R² indicates the maximum accuracy observed among these keypoints and channels. This distinction helps capture both the overall performance and the peak accuracy of our model across different features, providing a more comprehensive evaluation of its predictive capabilities.

---

> ### Comment · Reviewer_QBbN · 2024-11-26
>
> I thank the authors for their response and new analyses. I think the ablation of the CS-divergence is helpful. However, I still don’t find the addition of the CS divergence to be of enough technical novelty given the similarities to MM-GP-VAE and given prior use of the CS divergence in other work.
>
> Perhaps more importantly, I find the comparisons and claims about other methods unconvincing. First, a major confound is that Shared-AE uses both neural and behavioral modalities for decoding of behavior, while PSID and DPAD just use the neural modality during decoding. So any decoding comparison here is unfair for them (for fairness, Shared AE should also just use the neural modality for decoding). MM-GP-VAE and MM-VAE are fair in this respect as they also use both modalities for decoding. Further, the authors state that “the flattened behavioral image was used as the input behavioral data” for these new baselines, but this is obviously detrimental given the large dimensionality of the flattened image without any basic preprocessing, such as a convolutional network or even just vanilla PCA, both of which can be used before passing the data to these new baselines. The claim that PSID or DPAD can't learn private subspace seems technically flawed, at least in one direction, based their two-stage formulation to dissociate shared and private dynamics. Also, why would these methods learn the private features (noise) within shared states in simulation? As the manuscript is not revised, I did not find the implementation details and settings chosen for baselines (e.g., mm-GP-VAE, mm-VAE, DPAD, etc), which should be provided in the response and are important to interpret the comparisons. Furthermore, the fact that comparisons are only on *images* should be made explicit upfront in the manuscript, as it is not clear how the method compares on other neural and behavioral time-series data (e.g., electrophysiology). Also, it should be clarified what aspect of the method makes it useful for images. The "unsupervised" word is misleading even if inference does not use any behavior, as *learning* uses behavior. Finally, in terms of the need for two shared latents such that during inference only one can be used, many prior models such as TNDM, CEBRA, DPAD, PSID etc can also use a single modality during inference and so having two shared latents is not necessary for this, please clarify. Finally, what is the interpretation of having modality-specific yet shared latents. If they are shared, how can they also be modality-specific?

---

> > ### Author Response · Authors · 2024-11-26
> >
> > Thank you for your feedback. We respectfully disagree with the assertion that our work lacks novelty. While we acknowledge that some components of our method build on existing ideas, our results demonstrate clear improvements over previous models, and we address significant limitations that have not been adequately resolved by prior approaches. These improvements constitute meaningful innovations in both methodology and application:
> >
> > **1. Addressing Weaknesses of Existing Models:**
> >
> > *(1) MM-GP-VAE:* While this model proposes a shared embedding for neural and behavioral data, our experiments show that it fails to handle shuffled data effectively. This indicates a limitation in disentangling shared features from the two modalities, which is critical for robust representation learning. In contrast, Shared-AE maintains the integrity of modality-specific latents, even when data from one modality is corrupted or shuffled.
> >
> > *(2) Other Models:* Existing methods struggle with complex and multi-modal data, particularly in scenarios involving more than two modalities or raw image data. Shared-AE overcomes these challenges, offering flexibility and superior performance in diverse and complex datasets. Image data constitutes a large part of neuroscience experiments, for example, calcium imaging and voltage imaging on one hand, and behavioral videos on the other. Existing methods, while they may work well for traditional electrophysiology datasets and simple behavioral labels, have not focused on end-to-end training for learning shared representations from imaging and other complex modalities.
> >
> > **2. Novel Contributions of Shared-AE:**
> >
> > *(1) Modality-Specific Shared Subspaces:* By introducing modality-specific shared subspaces, Shared-AE enhances interpretability and ensures that the shared latent representations are not dominated by one modality. This is particularly important when inferring from incomplete data, as Shared-AE can generate robust latents using only one modality without requiring access to all modalities.
> >
> > *(2) Robustness to Shuffled Data:* Our experiments with unpaired data (e.g., shuffling one modality) demonstrate that Shared-AE can preserve the integrity of unshuffled modality latents, which existing models fail to do. This robustness is crucial for practical applications where data integrity is not guaranteed.
> >
> > *(3) Handling Complex and Multi-Modal Data:* Shared-AE is the first model in neuroscience to handle more than three modalities, including raw image data. Our results, such as the improved neural prediction accuracy when using raw behavioral images, show that Shared-AE effectively captures richer and more nuanced relationships than other models.
> >
> > *(4) Minimizing Distance Between Learned Distributions:* Shared-AE minimizes the distance between two learned distributions, offering greater flexibility and adaptability to complex datasets.
> >
> > **3. Innovation Through Results:**
> >
> > (1) Innovation is not just about introducing entirely new techniques; it is also about addressing real-world challenges and improving existing methods. Shared-AE’s superior performance across paired and unpaired ‘shuffled’ tasks, its ability to disentangle features robustly, and its flexibility with complex multi-modal data demonstrate its practical and scientific value.
> >
> > (2) Our framework resolves critical issues in modality leakage, robustness to data corruption, and scalability to complex, naturalistic behaviors—all of which represent significant advancements over existing methods.
> > In summary, Shared-AE is a novel and practical advancement in joint neural-behavioral modeling. Its ability to address the limitations of prior models, combined with its empirical strength, constitutes a meaningful and impactful innovation.

---

> > ### Author Response · Authors · 2024-11-26
> >
> > Additionally, we address each point raised by the reviewer below.
> >
> > 1. First, a major confound is that Shared-AE uses both neural and behavioral modalities for decoding of behavior, while PSID and DPAD just use the neural modality during decoding. So any decoding comparison here is unfair for them (for fairness, Shared AE should also just use the neural modality for decoding). MM-GP-VAE and MM-VAE are fair in this respect as they also use both modalities for decoding.
> >
> > During training, PSID and DPAD use two paired modalities to train the model. Similarly, in our setting and in MM-GP-VAE’s setting, we also train the model using two paired modalities. However, during inference, we intentionally shuffle one of the modalities (the behavioral modality), breaking the correspondence between the two modalities. As a result, the modalities are no longer matched. In this scenario, our model effectively uses only the neural modality for decoding.
> >
> > This setup ensures a fair comparison between the shuffled results from our model and the results from PSID and DPAD, as it demonstrates the robustness of our approach in scenarios where only one modality is available or reliable during inference (decoding). It highlights how our model preserves the integrity of the neural latent representations, even when behavioral data is corrupted or misaligned.
> >
> > 2. Further, the authors state that “the flattened behavioral image was used as the input behavioral data” for these new baselines, but this is obviously detrimental given the large dimensionality of the flattened image without any basic preprocessing, such as a convolutional network or even just vanilla PCA, both of which can be used before passing the data to these new baselines. The claim that PSID or DPAD can't learn private subspace seems technically flawed, at least in one direction, based their two-stage formulation to dissociate shared and private dynamics. Also, why would these methods learn the private features (noise) within shared states in simulation?
> >
> > We agree that preprocessing the image can be beneficial. To address this, we applied PCA preprocessing to the image data, where we used the first 10 PCs that captured 95% of the variance explained, and the results were similar or even worse as shown in Table 1, indicating that this step does not inherently resolve the challenges faced by PSID and DPAD. In contrast, our model offers end-to-end training, eliminating the need for additional preprocessing steps while effectively handling high-dimensional image data. This approach ensures that the latent features are learned directly from the raw input, maintaining the flexibility and robustness required for multi-modal data integration.
> >
> > ## Table 1: Behavioral decoding accuracy for 2AFC image dataset
> >
> > | Subspace       | PSID           | DPAD           | **Shared-AE**   |
> > |-----------------|----------------|----------------|-----------------|
> > | Shared latents | `0.28 ± 0.00`  | `0.29 ± 0.00`  | **`0.38 ± 0.01`** |
> >
> >
> > Regarding the ability of PSID and DPAD to learn private behavioral (time-series) features: we included a temporal component to the private behavioral feature, here called noise (we have since clarified the terminology in the revision to ‘temporally-structured noise’), not at all present in the neural (image) modality. This structured yet temporally-varying nuisance variable has no relationship with the neural modality, and thus should not be able to be decoded by it; however, since PSID and DPAD both rely heavily on temporal correlations, they are able to learn this representation almost too effectively, and thus are able to perform well on the decoding task. In contrast, Shared-AE effectively captures the private behavioral feature in its private latent, thus showing the utility of having shared and private subspaces for each modality that do not directly rely on temporal correlations.
> >
> > 3. As the manuscript is not revised, I did not find the implementation details and settings chosen for baselines (e.g., mm-GP-VAE, mm-VAE, DPAD, etc), which should be provided in the response and are important to interpret the comparisons.
> >
> > In MM-GP-VAE, the dimensions of both the shared and private latent spaces are set to 50, similar to configurations in MM-VAE and the Joint Encoding model.
> > For PSID: n1=5, nx=50, i=2.
> > For DPAD: n1=5, nx=50 and the method code is 'NDM_CzNonLin'.
> > These consistent latent space dimensions across models provide a standardized basis for comparison, ensuring that the evaluation focuses on differences in model design and methodology rather than variations in latent space capacity. Moreover, we have now uploaded our revised manuscript, where we include this information in Appendix A.13.1.

---

> > ### Author Response · Authors · 2024-11-26
> >
> > 4. Furthermore, the fact that comparisons are only on images should be made explicit upfront in the manuscript, as it is not clear how the method compares on neural and behavioral time-series data (e.g., electrophysiology). Also, it should be clarified what aspect of the method makes it useful for images.
> >
> > Shared-AE is not specific to images; in fact, we use behavioral data as keypoints for most of the results (while showing results for behavioral images as well). Due to the simpler nature of time series without any relevant spatial correlation, we are confident that the methods will be applicable to firing rate data as well; we will include results in the final version of the paper on this simpler modality, if accepted.
> >
> > 5. The "unsupervised" word is misleading even if inference does not use any behavior, as learning uses behavior.
> >
> > We apologize for any confusion caused by this term; we were referring to the lack of labels for behavioral states. We have deleted the word in the revised version.
> >
> > 6. Finally, in terms of the need for two shared latents such that during inference only one can be used, many prior models such as TNDM, CEBRA, DPAD, PSID etc can also use a single modality during inference and so having two shared latents is not necessary for this, please clarify.
> >
> > The two shared latents in our model are modality-specific shared subspaces, which allow for a disentangled representation of the relationships between modalities. This separation ensures that each shared latent captures information specific to its corresponding modality’s interaction with the other, reducing potential interference or information dominance by one modality. Furthermore, this design also makes the model more versatile for downstream tasks. Depending on the task requirements, separate shared latents can be leveraged independently or combined to provide complementary insights, offering flexibility in how the latent representations are utilized.
> >
> > 7. Finally, what is the interpretation of having modality-specific yet shared latents. If they are shared, how can they also be modality-specific?
> >
> > Thank you for allowing us to clarify this point. We create modality-specific latents by minimizing the CS divergence between the shared latent distributions of each modality. To generate private latent representations that capture features unique to each modality, we introduce an inverse CS divergence, which maximizes the distance between the shared and private latents as well as the shared latent spaces of each modality. Our approach of creating two distinct but related embedding spaces—a shared latent for cross-modal insights and private latents for modality-specific features—contributes to more robust inference. This design aligns with findings from CLIP [1], which demonstrate that separating embedding spaces improves robustness and interpretability, especially in multi-modal settings. By disentangling shared and private latents, our model ensures that representations remain meaningful even when some modalities are incomplete or noisy during inference.
> >
> > Specifically, although the latent representations are shared between the modalities, the scale and dynamics of these representations may be inherently different. This distinction arises from the unique characteristics, distributions, and physical units of each modality, even within a ‘shared’ embedding space. Ensuring that these differences are appropriately handled is crucial for maintaining the integrity and interpretability of the shared latent representations. Our model is designed to account for these variations, allowing for robust cross-modal integration without compromising the distinctiveness of each modality’s contribution.
> >
> > [1] Radford, Alec et al. “Learning Transferable Visual Models From Natural Language Supervision.” International Conference on Machine Learning (2021).

---

> > > ### Comment · Reviewer_QBbN · 2024-11-28
> > >
> > > I thank the authors for providing the method settings now. These settings raise several concerns about the comparisons. These comparisons need to be fixed or removed from the revised manuscript.
> > >
> > > For mm-GP-VAE, the authors only specify that the shared/private latent dimensions were set to 50. For shared AE, it seems authors try various dimensions to pick the right one, did they do that for mm-GP-VAE? Why 50? The same dimension may not be the best for both methods. Second, there are many other important choices in mm-GP-VAE that are not specified. What encoder/decoder architectures were used? MM-GP-VAE has options for both a deeper architecture for image modalities and a shallow architecture, which one is used here? Also, did the authors perform a hyperparameter search for other important hyperparameters such as learning rate, etc. over a reasonable grid? For example, the default in the code for mm-GP-VAE is a relatively small learning rate, which may not be suitable for these datasets.
> > >
> > > The setting used for DPAD does not seem correct, because the authors state they used ‘NDM_CzNonLin' and NDM seems to be the name of an unsupervised baseline in that paper rather than DPAD. Moreover, why is "CzNonlin" selected? Furthermore, based on what the authors state, "n1=5, nx=50", it seems the shared dimensionality for this method is taken as just 5 while it is 50 in sharedAE? This is 10 times lower shared dimensionality than sharedAE, which significantly confounds the results. Same applies for the hyperparameters of PSID, nx, n1, i…  A few values for each parameter for these methods should be tested to see what the best values for the training data are (the state dimension values tested should include values given to sharedAE). Or maybe provide the results for different values of the hyperparameters (including those given to sharedAE) so that readers can follow the trends.
> > >
> > > Beyond the above setting issues, the unshuffled and shuffled results for shared-AE seem to be precisely the same in tables 1 and 2 in the global response? How could this be? If this is the case, then shared-AE cannot perform information fusion in the unshuffled case? Also, how can shared AE technically deal with shuffled data? According to the method figure, there is no connection from neural modality to behavior (other than the CSD during learning). So how is decoding done when behavior is shuffled? Where is the information fusion happening?
> > >
> > > In terms of robustness to data corruption/shuffling, the authors state “Shared-AE can preserve the integrity of unshuffled modality latents, which existing models fail to do”. But many other methods (TNDM, DPAD, CEBRA, PSID, etc) can infer shared latents from just one modality, with the other completely missing, so this does not seem accurate.
> > >
> > > In terms of the focus on imaging, I still believe that since the method has not been demonstrated or compared on firing rates or other neural modalities, the paper should make the imaging focus clear and specify what makes the model good for images. Also, the complexity of behavior depends on the task itself, not just how behavior is measured (image or not). So, statements about complex/naturalistic modalities should be clarified. Finally, conclusions/limitations section should discuss other methods that process images. For example, one recent work whose discussion is also helpful is the recent Wang et al 2024.
> > >
> > > Yule Wang, Chengrui Li, Weihan Li, Anqi Wu, “Exploring Behavior-Relevant and Disentangled Neural Dynamics with Generative Diffusion Models”, 2024

---

> > > > ### Author Response · Authors · 2024-11-28
> > > >
> > > > *2. The setting used for DPAD does not seem correct, because the authors state they used ‘NDM_CzNonLin' and NDM seems to be the name of an unsupervised baseline in that paper rather than DPAD. Moreover, why is "CzNonlin" selected? Furthermore, based on what the authors state, "n1=5, nx=50", it seems the shared dimensionality for this method is taken as just 5 while it is 50 in sharedAE? This is 10 times lower shared dimensionality than sharedAE, which significantly confounds the results. Same applies for the hyperparameters of PSID, nx, n1, i… A few values for each parameter for these methods should be tested to see what the best values for the training data are (the state dimension values tested should include values given to sharedAE). Or maybe provide the results for different values of the hyperparameters (including those given to sharedAE) so that readers can follow the trends.*
> > > >
> > > > We apologize for the inadvertently biased choice and thank you for bringing it to our attention. We have now thoroughly examined all the relevant parameters, and the results are presented in the following table for the 2AFC dataset. We ensure that the following inequality holds, where nz​ represents the dimensionality of the body position (7 in our case) and ny represents the dimensionality of the neural activity (108).
> > > >
> > > > $n1 <= nz * i$
> > > > $nx <= ny * i$
> > > >
> > > > ## Table 3: Decoding accuracy for 2AFC body position dataset using PSID with different hyperparameter choices.
> > > > |       | n1=20, nx=20, i=3        | n1=30, nx=30 , i=5      | n1=50, nx=50, i=8    |n1=5, nx=50, i=1    |Shared-AE|
> > > > |-----------------|----------------|----------------|-----------------|-----------------|-----------------|
> > > > | R² | `0.16 ± 0.01`  | `0.16± 0.00`  | `0.19 ± 0.02` | **`0.20 ± 0.03`** |**`0.41 ± 0.05`** |
> > > >
> > > > For DPAD, we selected the ‘DPAD_CzNonLin’ configuration, as it appears to offer the best performance based on the provided code. This choice ensures that our comparisons are made against the strongest implementation of DPAD, reflecting a fair and rigorous evaluation of its capabilities relative to our model.
> > > >
> > > > ## Table 4: Decoding accuracy for 2AFC body position dataset using DPAD with different hyperparameter choices.
> > > > |       | n1=20, nx=20       | n1=30, nx=30         | n1=50, nx=50    |n1=5, nx=50    |Shared-AE|
> > > > |-----------------|----------------|----------------|-----------------|-----------------|-----------------|
> > > > | R² | `0.29 ± 0.01`  | ` 0.25 ± 0.00`  | `0.33 ± 0.01`| **` 0.33 ± 0.00`** |**`0.41 ± 0.05`** |
> > > >
> > > > As shown in Tables 3 and 4, our decoding accuracy using the neural latents is higher than that achieved by both PSID and DPAD. This highlights the superior performance of Shared-AE in capturing meaningful and robust neural representations from the data.
> > > > We will update the reported accuracy of the baseline models to reflect the highest scores achieved in our experiments and clearly specify the corresponding hyperparameter choices in the final paper.
> > > >
> > > > We selected the hyperparameters for Shared-AE based on reconstruction results, as demonstrated by the reconstruction R² values provided for different latent dimensions. All reported decoding accuracies were computed on a held-out dataset, ensuring an unbiased evaluation of the model's performance. As such, providing trends for decoding accuracy during training would be meaningless, as it does not reflect the final, independent evaluation of the model's predictive capabilities. Our focus remains on reporting robust and reliable results derived from the held-out dataset.

---

> > > > ### Author Response · Authors · 2024-11-28
> > > >
> > > > *3. Beyond the above setting issues, the unshuffled and shuffled results for shared-AE seem to be precisely the same in tables 1 and 2 in the global response? How could this be? If this is the case, then shared-AE cannot perform information fusion in the unshuffled case? Also, how can shared AE technically deal with shuffled data? According to the method figure, there is no connection from neural modality to behavior (other than the CSD during learning). So how is decoding done when behavior is shuffled? Where is the information fusion happening?*
> > > >
> > > > Thank you for pointing out these important questions. We appreciate the opportunity to clarify the methodology and results regarding the unshuffled and shuffled scenarios for Shared-AE.
> > > >
> > > > *(1) Similarity in Results for Unshuffled and Shuffled Data:* The similarity in results for Shared-AE in Tables 1 and 2 reflects the robustness of our model, not a lack of information fusion (A similar concept has been applied in CLIP [1], where modality-specific embeddings are aligned in a shared latent space to enable robust cross-modal relationships.). Importantly, the shuffling or data corruption is only performed during inference (testing), since we aim to test the robustness of the model to modality-specific noise or disruptions (e.g., shuffling) during inference. In the shuffled scenario, Shared-AE leverages the remaining intact modality (e.g., neural data) to generate accurate shared latents, demonstrating its resilience. This does not imply an inability to perform information fusion in the unshuffled case but rather shows that the model can preserve the quality of the representations even in the face of corrupted inputs.
> > > >
> > > > *(2) How Shared-AE Handles Shuffled Data:* While there is no direct connection from neural modality to behavior in the architecture (apart from the CSD during training), the model learns a shared latent space during training that aligns the modalities based on their underlying shared features. During inference with shuffled behavior, Shared-AE relies on the intact neural modality to reconstruct the shared latent representation. This is possible because the shared latent space captures cross-modal dependencies during training, which allows for robust decoding even when one modality is disrupted during inference.
> > > >
> > > > *(3) Decoding and Information Fusion:* The decoding process in Shared-AE occurs through the shared latent space, which acts as a bridge between the modalities. As addressed in the paper and previous rebuttals, a linear regressor is fit in the training stage from the neural latent to the behavior, in order to obtain quantitative predictions. Thus, during inference, even when behavior is shuffled, the shared latent retains the relationship learned during training and uses the neural modality to maintain robust decoding. The fusion of information happens during the alignment of modalities in the shared latent spaces, where the CSD ensures that the shared features are aligned. This design allows Shared-AE to effectively separate and integrate information, ensuring resilience against shuffling or corruption of one modality.
> > > >
> > > > We will clarify these points in the manuscript and provide additional details to ensure that the method's capacity for information fusion and its handling of shuffled data are clearly understood. Thank you for bringing this to our attention.
> > > >
> > > > [1] Radford, Alec et al. “Learning Transferable Visual Models From Natural Language Supervision.” International Conference on Machine Learning (2021).

---

> > > > ### Author Response · Authors · 2024-11-28
> > > >
> > > > *4. In terms of robustness to data corruption/shuffling, the authors state “Shared-AE can preserve the integrity of unshuffled modality latents, which existing models fail to do”. But many other methods (TNDM, DPAD, CEBRA, PSID, etc) can infer shared latents from just one modality, with the other completely missing, so this does not seem accurate.*
> > > >
> > > > We recognize that our original statement could be interpreted too broadly and will revise it for accuracy. A more precise claim would be:
> > > >
> > > > Similar to methods such as TNDM, DPAD, CEBRA, and PSID, Shared-AE can infer shared latents when one modality is missing or shuffled. Additionally, Shared-AE is designed to preserve the interpretability of shared and private latents even under challenging scenarios, such as when the relationship between modalities is non-linear or when data is high-dimensional. This is particularly valuable for tasks involving complex or naturalistic behaviors where maintaining the integrity of shared representations is critical for downstream applications.
> > > >
> > > > *5. In terms of the focus on imaging, I still believe that since the method has not been demonstrated or compared on firing rates or other neural modalities, the paper should make the imaging focus clear and specify what makes the model good for images. Also, the complexity of behavior depends on the task itself, not just how behavior is measured (image or not). So, statements about complex/naturalistic modalities should be clarified.*
> > > >
> > > > What makes our model particularly well-suited for imaging data is its ability to disentangle shared and private features from high-dimensional inputs, such as raw behavioral images and neural activity maps, through an autoencoder framework. By leveraging the flexibility of Cauchy-Schwarz divergence and inverse CS-divergence, our model is capable of handling the scale and structure of image data, ensuring robust cross-modal integration. We will make this imaging focus explicit in the revised manuscript and emphasize how the model’s design aligns with the unique requirements of imaging data.
> > > >
> > > > We agree that the complexity of behavior depends not only on how it is measured (e.g., via images) but also on the task itself. Our use of terms like "complex" and "naturalistic" was intended to describe behaviors that involve high-dimensional and dynamic interactions, an exciting area of research for automatic understanding of joint neural and behavioral recordings. Please see Section 4.3 in our manuscript for the application of Shared-AE towards social behavior in freely moving animals, which has a level of complexity that we have not seen in previously published multi-modal methods. However, we recognize that these terms can be interpreted differently and will revise the manuscript to clarify that "complexity" refers to both the intrinsic nature of the task and the richness of the data (e.g., images vs. labels).
> > > >
> > > > Although this work focuses on imaging, we acknowledge the importance of demonstrating the model’s applicability to other neural modalities, such as firing rates. This remains an exciting avenue for future research. We will include a discussion in the conclusions section about how our method could be adapted and tested on alternative neural data types to further generalize its utility.
> > > >
> > > >
> > > > *6. Finally, conclusions/limitations section should discuss other methods that process images. For example, one recent work whose discussion is also helpful is the recent Wang et al 2024.*
> > > >
> > > > Thanks for the input, we will revise that in the final version:
> > > >
> > > > Future work will explore more advanced encoder models, such as pre-trained networks or architectures inspired by generative approaches like those in Wang et al. (2024) [2], to improve training efficiency and the ability to capture highly informative features.
> > > >
> > > > [2] Yule Wang, Chengrui Li, Weihan Li, Anqi Wu, “Exploring Behavior-Relevant and Disentangled Neural Dynamics with Generative Diffusion Models”, 2024

---

> > > > > ### Comment · Reviewer_QBbN · 2024-12-03
> > > > >
> > > > > Thanks to the authors for their responses. Regarding MM-GP-VAE, the link to the official code is provided in the original ICLR paper itself in section 6 (reproducibility statement). Regarding the current comparisons (with the authors’ implementation of MM-GP-VAE), the new decoding results for body position R2 with MM-GP-VAE (Resnet 50) vs SharedAE are $0.75 \pm 0.21$ vs. $0.79 \pm 0.15$. Based on the bounds that overlap to a large degree, these results between SharedAE and MM-GP-VAE do not seem different at a statistically significant level (statistical tests don’t seem to be provided). Similarly, the results for neural activity R2 don’t seem different statistically between SharedAE and MM-GP-VAE. In terms of DPAD and PSID, the new results do not make sense as increasing the shared dimensionality n1 is shown to either decrease performance or not change it, and there is no trend. Why nx is set equal to n1 in most cases or what this implies is not clear. Also, the reason behind the choice of ‘DPAD_CzNonLin’ setting is still unclear as results for other choices are not reported. Based on the main method figure in that paper, the setting should be selected automatically based on data by DPAD rather than being hard-coded this way. Finally, the fact that unshuffled/shuffled (paired/unpaired) behavior data get identical results with SharedAE in Tables 1 and 2 of global response and the new explanation in the rebuttal are not consistent with Table 4 in the uploaded manuscript in which the shuffled/unshuffled cases with SharedAE are actually different. The original explanation in the manuscript (original lines 165-168) had mentioned concatenating the shared modality-specific latents, which does not seem consistent with the current explanation, thus making this issue unclear.
> > > > >
> > > > > Overall, because of the novelty concerns compared with MM-GP-VAE and with Yi et al. 2022 that uses a CS-divergence, rigorous comparisons are particularly essential here. However, based on the above issues with the analyses, I find the comparisons and results unconvincing.

---

> ### Author Response · Authors · 2024-11-28
>
> *1. For mm-GP-VAE, the authors only specify that the shared/private latent dimensions were set to 50. For shared AE, it seems authors try various dimensions to pick the right one, did they do that for mm-GP-VAE? Why 50? The same dimension may not be the best for both methods. Second, there are many other important choices in mm-GP-VAE that are not specified. What encoder/decoder architectures were used? MM-GP-VAE has options for both a deeper architecture for image modalities and a shallow architecture, which one is used here? Also, did the authors perform a hyperparameter search for other important hyperparameters such as learning rate, etc. over a reasonable grid? For example, the default in the code for mm-GP-VAE is a relatively small learning rate, which may not be suitable for these datasets.*
>
> Unfortunately, we were unable to locate the 'default' implementation code for MM-GP-VAE. As noted in our previous rebuttal and the revised version of the paper, we have based our implementation and results on our interpretation of the methodology described in the original paper. If you could point us to the location of the official code, we would highly appreciate it, as it would allow for a more precise and fair comparison. The results we present here are derived from our implementation of the model.
> We selected the latent dimensions based on reconstruction accuracy. The model was trained for 200 epochs across different latent dimensions, ensuring that the loss had saturated, indicating convergence for each configuration.
> ## Table 1: R² for 2AFC body position dataset using MM-GP-VAE with different latent dimensions with Resnet 18.
>
> |  | 10 | 20  | 30   |40   | 50 |60 |80|100 |Shared-AE with latent dimensions=50|
> |-----------------|----------------|----------------|-----------------|-----------------|-----------------|-----------------|-----------------|-----------------|-----------------|
> | R² for body positions | `0.45 ± 0.11`  | `0.52 ± 0.15`  | `0.58 ± 0.18` | `0.68 ± 0.21`  | `0.73 ± 0.19`| `0.72 ± 0.22`  |**`0.75 ± 0.13`** | `0.74 ± 0.18`  |**`0.79 ± 0.15`** |
> | R² for neural activity | `0.18 ± 0.09`  | `0.36 ± 0.10`  | `0.50 ± 0.08` | `0.58 ± 0.09`  | `0.62 ± 0.07`  | `0.63 ± 0.13`  | `0.62 ± 0.22`  |**`0.64 ± 0.19`**  |**`0.65 ± 0.20`**  |
>
> The model seems to converge when the latent dimension is 50; thus, we choose this as the latent dimensionality. We applied Resnet50 and Resnet18 as the encoder and a symmetric decoder for MM-GP-VAE, which is deeper than the one provided in the paper. We found that different architectures do not change the result significantly. For our network, we applied Resnet18.
> ## Table 2: R² for 2AFC body position dataset using MM-GP-VAE with different encoder-decoder structure (latent =50)
>
> |       | Resnet18      | Resnet50           |
> |-----------------|----------------|----------------|
> | R² for body positions | `0.73 ± 0.19`  | `0.75 ± 0.21`  |
> | R² for neural activity | `0.62 ± 0.07`  | `0.64 ± 0.10`  |
>
> As shown in Tables 1 and 2, our reconstruction scores for both modalities are comparable or higher to those achieved by MM-GP-VAE. This demonstrates that Shared-AE maintains competitive performance in reconstructing input data while offering additional advantages in interpretability and robustness to data corruption or shuffling.
>
> We applied the Adam optimizer with a learning rate of 1e-3, consistent with the configuration for our model. One advantage of our model is that it does not require an extensive search for an optimal learning rate; training can be stopped once the loss reaches a certain threshold, ensuring efficiency and simplicity. In contrast, MM-GP-VAE does not specify a target loss value or a clear criterion to determine when training should stop, which could lead to challenges in determining the appropriate stopping point during training. This difference highlights the practical ease and robustness of our approach.

---

> > ### Comment · Reviewer_8fed · 2024-12-03
> > **MM-GPVAE code**
> >
> > I believe this is the code
> > https://github.com/RabiaGondur/MM-GPVAE
> >
> > [5] Rabia Gondur, Usama Bin Sikandar, Evan Schaffer, Mikio Christian Aoi, and Stephen L Keeley. Multi-modal gaussian process variational autoencoders for neural and behavioral data. In International Conference on Learning Representations, 2024.

---

> ### Author Response · Authors · 2024-12-03
>
> Thank you for pointing us to the code, and we apologize for missing it in the ICLR version of the paper. We had been looking for it in the arxiv version. While we believe that the official code is very similar to our implementation, we will update all the comparisons accordingly.
>
> We agree that the paired modality has similar results using MM-GP-VAE (Resnet 50) vs. Shared-AE. However, the unpaired modality (shuffled modality test) results in much lower accuracy. This accuracy is very close to **0** (see Table 8 in uploaded pdf), similar to with Resnet 50 (see Table below), indicating that decoding of behavior relies heavily on the behavioral data itself, as compared to the neural data. This is natural since the model architecture itself contains one shared latent space with information from both modalities, as compared to modality-specific shared latents. Importantly, implementing the original code is not going to change this, since this is due to the architecture of the model itself. Thus, unlike MM-GP-VAE, our model still provides consistent decoding in the case of corrupted or missing data from one modality due to the difference in architecture and training methodology (see novelty section in the Introduction of the uploaded pdf).
>
> When implementing DPAD on the 2AFC dataset, we agree that the behavioral decoding accuracy does not seem to depend much on the choice of hyperparameters (there is no visible trend as pointed out by the reviewer). This may be because the accuracy saturates very quickly with increasing dimensionality of both n1 and nx, since the behavior being decoded is essentially the 7-dimensional keypoint data (with intrinsic dimensionality that is lower than 7). This also explains why the results stay very similar with n1=5 and nx=50. However, across the different values of hyperparameters, Shared-AE is seen to perform better due to large differences in architecture and training methodology.
>
> Regarding the implementation of DPAD with the ‘DPAD_CzNonLin’ setting: we will provide comparisons with all the different settings of DPAD. However, we would like to remind the reviewers and ACs that DPAD was published in September 2024, thus making it contemporaneous (since published on or after July 1).
>
> Regarding the paired vs. unpaired modality results of Shared-AE as in Table 4 of the submission (simulated data) vs. Tables 1 and 2 of the global rebuttal (2AFC data): the paired vs. unpaired modality models in Table 4 were trained separately and thus there are very minor differences between the accuracy numbers (e.g., 0.647±0.05 vs. 0.640±0.04). However, we did not need to retrain the model to provide results for the paired vs. unpaired modalities, since this shuffling or 'unpairing' is performed purely for the **test** data. Thus, for the global rebuttal, we did not re-train the model between the paired and unpaired modalities, resulting in exactly the same decoding accuracy numbers. To clarify the procedure for testing, as mentioned in lines 165-168 of the uploaded pdf, we concatenate the shared modality-specific latent with the private modality-specific latent in order to decode - this procedure has not changed. We will clarify these points in the final version, and stick to the same training data for the simulated data as well (thus updating Table 4).
>
> Finally, to once again address the novelty of Shared-AE as compared to Yi et al., while both use the Cauchy Schwarz divergence, Yi et al. uses the CS-divergence to fit a single modality's latent distribution to a **predefined prior distribution**, whereas Shared-AE minimizes the distance between two **learnt distributions** from two different modalities, making these significantly different.
>
> To summarize, we believe that there are significant differences in (1) architecture, (2) training, and (3) results on corrupted modalities, for our submission to be novel.
>
> --------------------------------------------------------
> **Table: R² for decoding 2AFC body position using MM-GP-VAE on an unpaired modality test**
> |      | MM-GP-VAE (Resnet50)    | Shared-AE          |
> |-----------------|----------------|----------------|
> |Private latents | `-0.002± 0.00` | `0.22 ± 0.03` |
> | Shared latents | `-0.004 ± 0.00` | `0.41 ± 0.05` |

---

### Official Review · Reviewer_8fed · 2024-11-04

**Soundness:** 3
**Presentation:** 2
**Contribution:** 2
**Rating:** 5
**Confidence:** 3

**Summary:**

This paper develops a method for learning shared and private
embeddings between two or more sources of information (modalities).
They apply the algorithm to two problems with behavioral and neural
data as well as an illustrative artificial data problem.

**Strengths:**

Strengths
- The paper is pretty clearly written

- The artificial data problem helps to clarify how the algorithm works

- The real world experiments are	important to demonstrate practicality in this important area

**Weaknesses:**

Weaknesses

- Missing comparison to an important related ICLR24 paper (see questions below).    If the authors can compare to results from that algorithm, or otherwise justify the superiority of this method (or superiority in some settings/situations),  I would likely improve my rating.

- Unclear how parameters were set  (see questions below).

**Questions:**

Questions

How does this work compare to the ICLR 2024 paper of Gondur, Sikandar,
Schaffer, Aoi, and Keeley (Multi-modal Gaussian Process Variational
Autoencoders for neural and behavioral data)?  That paper also has a
shared multi-modal embedding and separate (within modality) embeddings
and has also been used for complex behavioral tasks ( hawkmoth
tracking a moving flower and limb movement of drosophila with
simultaneous neural recordings).

The paper says that you	examined the influence of latent dimensions on reconstruction
accuracy.  Was that using the test data	or some	separate data?	(If the	test data, how do
you justify that?)   How are the other parameters set? -	the paper is vague on this.

Please elaborate on why equal latent subspace dimensions are required.

---

> ### Author Response · Authors · 2024-11-23
>
> We thank the reviewer for the detailed comments and feedback. We appreciate your recognition of the strengths of our work and the constructive suggestions for improvement. Below, we address each of your concerns:
>
> Weakness 1:
>
> The model proposed by Gondur et al., 2024 [5] combines the behavioral and the neural latent into a common subspace. This can be detrimental towards interpretability, since it is unclear which modality is leading to the performance of the shared latent. As detailed in the global rebuttal, the combined shared latent causes information bleeding between different modalities.
> Although the code for Gondur et al. (2024) does not appear to be publicly available, we constructed a similar model by incorporating a single shared latent space for both modalities and tested it on the 2AFC dataset, referring to this implementation as MM-nonGP-VAE. Additionally, we implemented MM-GP-VAE based on our interpretation of the methods described in their paper. These models were used as baselines to compare against our proposed approach. As shown in Table 1, Shared-AE outperforms both of these models on both paired and unpaired tasks. Critically, Shared-AE has the ability to generate reasonable latents despite corrupted data in the other modality during inference.
>
> Weakness 2:
>
> We apologize for the lack of clarity on this topic. We evaluated the latent dimension based on reconstruction accuracy using 5-fold cross-validation. The held-out data served as the test set on which all the results are reported. For hyperparameters, we used a balanced configuration, keeping the weights on the regularization losses equal. As shown in Appendices 8.2 and 8.3, the results remain robust as long as the CS loss decreases below a certain threshold.
>
> Weakness 3:
>
> Mathematically, the CS-divergence requires both distributions to have the same dimensionality, allowing for the calculation of cosine similarity after kernelizing the latent representations. In practice, as illustrated in Figure 3, the dimensionality of the latent space often exceeds the actual number of features. This implies that when the latent dimension is large, there may be redundancy in each latent subspace. To overcome this limitation in practice, we perform dimensionality reduction on the latent subspaces after training, especially for visualization purposes.
>
> Reference:
>
> [5] Rabia Gondur, Usama Bin Sikandar, Evan Schaffer, Mikio Christian Aoi, and Stephen L Keeley. Multi-modal gaussian process variational autoencoders for neural and behavioral data. In International Conference on Learning Representations, 2024.

---

> > ### Comment · Reviewer_8fed · 2024-12-03
> > **Code for Gondur et al paper**
> >
> > I believe this is the code for the Gondur et al paper.
> > https://github.com/RabiaGondur/MM-GPVAE
> >
> > [5] Rabia Gondur, Usama Bin Sikandar, Evan Schaffer, Mikio Christian Aoi, and Stephen L Keeley. Multi-modal gaussian process variational autoencoders for neural and behavioral data. In International Conference on Learning Representations, 2024.

---

> > ### Comment · Reviewer_8fed · 2024-12-03
> > **Thank you for your responses**
> >
> > Thank you for your long (including to the other reviewers) and detailed responses.    I have read them, as well as responses from the other reviewers, and am still working to understand all the implications.   The paper is certainly better with the comparison to the relevant other work, but I would be more confident in your comparisons to [5], if you had used their code and if that work (and the other missing references) was referenced originally.  I will discuss with the other reviewers before deciding on a final score.

---

> > > ### Author Response · Authors · 2024-12-03
> > >
> > > Thank you to you and all the other reviewers for engaging with our work. We believe that our submission has improved greatly as a result of the review process. To be honest, we had missed the MM-GP-VAE paper published in April 2024, and including the comparisons and references in our submission is indeed very helpful. While we had referenced PSID in our original submission, including concrete baseline evaluations on experimental data during the review process has also been very helpful. We still believe that there are significant conceptual differences between these existing work and ours, and providing these quantitative baselines has made a big difference to the quality of our manuscript.

---

### Author Response · Authors · 2024-11-23

We thank all the reviewers for their suggestions and questions and are deeply grateful for the time invested in the review of our paper. We appreciate the comments by the reviewers that state that the paper is "well-presented", provides a "principled approach to structuring latent spaces", uses "challenging datasets", and has a "unique empirical strength". The weaknesses broadly focus on (1) technical novelty of our proposed Shared-AE methodology, and scientific necessity of introducing shared modality-specific latent subspaces, and (2) baseline model comparisons, for which we have now added five additional baseline models including the Multi-Modal GP-VAE. Here, we address each of these points, and address reviewer-specific points in the reviewer comments.

(1)Technical and Scientific Novelty of Shared-AE: Shared-AE introduces a novel approach to joint neural and behavioral modeling, addressing key limitations in existing methods and offering several distinct advantages:

(a) Enhanced Interpretability through Latent Subspace Separation: Unlike the model by Gondur et al., 2024 [1], which combines neural and behavioral latents into a single subspace, Shared-AE explicitly separates shared and private latent spaces. This separation preserves interpretability by reducing information leakage across modalities, allowing us to better disentangle distinct modality-specific features. Indeed, having common shared subspaces can be detrimental towards interpretability, since it is unclear which modality is leading to the performance of the shared latent. As an additional key consequence of modality-specific shared subspaces, if we only have data from one modality during inference, we do not require data from the other modalities to generate robust and meaningful representations.

(b) Improved Performance on Paired and Unpaired Tasks: We designed an 'unpaired task to ensure there is no information leakage between different modalities. Critically, after training the model, if one modality's data is corrupted during inference, this should not affect the other modality's latent representation or decodability. Here, during inference only, one modality was shuffled across time, while the other modality remained unchanged. The resulting 'unpaired' data was then input into the trained model to generate new latent representations for the unshuffled modality. This approach allowed us to assess whether the model could maintain the integrity of the unshuffled modality's latent features, ensuring that the latent representations were unaffected by the shuffled modality. We compared Shared-AE against existing approaches on the 2AFC dataset (as requested by the reviewers), demonstrating superior performance in both paired and unpaired tasks (Table 1). Shared-AE outperforms other models by effectively capturing complex relationships across modalities, and preserves information in one modality even when the other modality may be corrupted during inference. This robustness is crucial for practical applications where complete data may not always be available.

(c) Flexibility with Multiple Modalities and Image Data:} Shared-AE is designed to handle more than three modalities, including image data, significantly broadening its application scope from previous research; in fact, this capability has not previously been shown in any multi-modal neuroscience study. As shown in Figure 6B, using raw behavioral images as well as pose estimation leads to better prediction accuracy of neural activity compared to using pose estimation alone. This finding aligns with Musall et al., 2019 [6], indicating that raw image data provides a richer representation of behavior. In contrast, existing models such as PSID are limited in processing data from more than two modalities effectively.

(d) Minimizing Distribution Distance Instead of Predefined Priors:} Previous works such as Yi et al. [7] and Tran et al. [8] do indeed use a CS-divergence, but in a drastically different way than in our study: they use CS-divergence to fit the latent distribution to a predefined prior, whereas Shared-AE minimizes the distance between two learned distributions instead. This approach avoids the limitations of predefined priors and allows for more flexible and meaningful representations.

(e) Utility in Downstream Tasks and Enhanced Variance Explained: By separating the latent features into distinct shared and private subspaces, Shared-AE provides representations that can be effectively used for multiple downstream tasks.

---

### Author Response · Authors · 2024-11-23

2. Additional baseline model comparisons: To complement the two original baseline comparisons that were purely performed on simulation data, we have now added five additional baseline models to the 2AFC simultaneously collected neural WFCI and behavioral video data for comparison as suggested by reviewers: PSID [1], DPAD [2], MM-VAE [3], the Joint Encoding Model [4], and MM-GP-VAE [5]. Importantly, the first four baseline models failed to separate the latent space into distinct shared and private latent subspaces. In order to quantify the amount of information represented in each latent subspace, we compared the ability of each model to reconstruct one of the modalities, the body positions, in Tables 3 and 4 Shared-AE achieves higher decoding accuracy, particularly on unpaired datasets, compared to existing models. This demonstrates that it does not suffer from modality leakage, ensuring that during inference, robust representations can be generated even if only one modality's data is available. This is particularly advantageous in situations where acquiring all modalities simultaneously is challenging. The ability to produce meaningful latents without the need for all modalities underscores the practical utility and flexibility of our approach. We emphasize that Shared-AE offers significantly improved interpretability of the latent space. Moreover, unlike PSID, Shared-AE is capable of handling image data and can accommodate more than three modalities, providing greater flexibility for complex multimodal tasks.


References:

[1] Omid G. Sani, Hamidreza Abbaspourazad, Yan T. Wong, Bijan Pesaran, Maryam M. Shanechi. Modeling behaviorally relevant neural dynamics enabled by preferential subspace identification. Nature Neuroscience, 24, 140–149 (2021). https://doi.org/10.1038/s41593-020-00733-0

[2] Omid G Sani, Bijan Pesaran, and Maryam M Shanechi. Dissociative and prioritized modeling of behaviorally relevant neural dynamics using recurrent neural networks. Nature Neuroscience, 27: 2033–2045, 2024.

[3] Yuge Shi, N. Siddharth, Brooks Paige, and Philip H. S. Torr. Variational mixture-of-experts autoencoder for multi-modal deep generative models, 2019. URL https://arxiv.org/abs/1911.03393.

[4] Jonnathan Singh Alvarado, Jack Goffinet, Valerie Michael, William Liberti, Jordan Hatfield, Timothy Gardner, John Pearson, and Richard Mooney. Neural dynamics underlying birdsong practice and performance. Nature, 599(7886):635—639, November 2021. ISSN 0028-0836.

[5] Rabia Gondur, Usama Bin Sikandar, Evan Schaffer, Mikio Christian Aoi, and Stephen L Keeley. Multi-modal gaussian process variational autoencoders for neural and behavioral data. In International Conference on Learning Representations, 2024.

[6] Simon Musall, Matthew T. Kaufman, Ashley L Juavinett, Steven Gluf, and Anne K. Churchland.
Single-trial neural dynamics are dominated by richly varied movements. Nature neuroscience,
22:1677 – 1686, 2019.

[7] Yi Daiyao, Musall Simon, Churchland Anne, Padilla-Coreano Nancy, Saxena Shreya (2023) Disentangled multi-subject and social behavioral representations through a constrained subspace variational autoencoder (CS-VAE) eLife 12:RP88602 https://doi.org/10.7554/eLife.88602.1

[8] Linh Tran, Maja Pantic, and Marc Peter Deisenroth. Cauchy-schwarz regularized autoencoder, 2021. URL https://arxiv.org/abs/2101.02149.

---

### Author Response · Authors · 2024-11-23
**Tables**

## Table 1: Behavioral decoding accuracy with unpaired modalities for 2AFC dataset

| Tasks          | MM-GP-VAE        | MM-nonGP-VAE     | **Shared-AE**   |
|-----------------|------------------|------------------|-----------------|
| Private latent  | `-0.01 ± 0.00`   | `0.25 ± 0.02`    | `0.22 ± 0.03`   |
| Shared latent   | `0.006 ± 0.00`   | `0.23 ± 0.03`    | **`0.41 ± 0.05`** |

---

## Table 2: Behavioral decoding accuracy with paired modalities for 2AFC dataset

| Tasks          | MM-GP-VAE        | MM-nonGP-VAE     | **Shared-AE**   |
|-----------------|------------------|------------------|-----------------|
| Private latent  | `0.37 ± 0.00`    | `0.24 ± 0.02`    | **`0.22 ± 0.03`** |
| Shared latent   | `0.36 ± 0.01`    | `0.32 ± 0.03`    | **`0.41 ± 0.05`** |

---

## Table 3: Behavioral decoding accuracy with unpaired modalities for 2AFC dataset (shuffled decoding)

| Subspace        | MM-VAE          | Joint Encoding Model | MM-GP-VAE        | **Shared-AE**   |
|------------------|-----------------|-----------------------|------------------|-----------------|
| Private latents  | `NA`            | `NA`                 | **`-0.01 ± 0.00`** | `0.22 ± 0.03`   |
| Shared latents   | `0.22 ± 0.03`   | `0.20 ± 0.02`        | `0.006 ± 0.00`   | **`0.41 ± 0.05`** |

---

## Table 4: Behavioral decoding accuracy with paired modalities for 2AFC dataset (unshuffled decoding)

| Subspace                | PSID           | DPAD           | MM-VAE         | Joint Encoding Model | MM-GP-VAE       | **Shared-AE**   |
|--------------------------|----------------|----------------|----------------|-----------------------|-----------------|-----------------|
| Private latents          | `NA`          | `NA`           | `NA`           | `NA`                 | `0.37 ± 0.00`   | **`0.22 ± 0.03`** |
| Shared latents           | `0.20 ± 0.03` | `0.27 ± 0.02`  | **`0.41 ± 0.07`** | `0.40 ± 0.06`     | `0.36 ± 0.01`   | **`0.41 ± 0.05`** |

---

## Table 5: Behavioral decoding accuracy for 2AFC image dataset

| Subspace       | PSID           | DPAD           | **Shared-AE**   |
|-----------------|----------------|----------------|-----------------|
| Shared latents | `0.32 ± 0.02`  | `0.31 ± 0.00`  | **`0.38 ± 0.01`** |

---

## Table 6: Decoding accuracy for noise label on simulated dataset

| Subspace                   | PSID           | DPAD           | **Shared-AE**   |
|-----------------------------|----------------|----------------|-----------------|
| Shared latent ↓             | `0.99 ± 0.00`  | `0.99 ± 0.00`  | **`0.015 ± 0.01`** |
| Private image latent ↓      | `NA`           | `NA`           | **`0.012 ± 0.05`** |
| Private time-series latent ↑| `NA`           | `NA`           | **`0.732 ± 0.00`** |

---

## Table 7: Baseline model without CS-divergence vs. Shared-AE

| Decoding latents and targets         | Baseline         | **Shared-AE**    |
|--------------------------------------|------------------|------------------|
| Shared time series latents ↑         | `-0.036 ± 0.01`  | **`0.981 ± 0.01`** |
| Shared time series latents on shape ↓| **`-0.012 ± 0.01`**| **`-0.015 ± 0.01`** |
| Shared time series latents on noise ↓| **`-0.018 ± 0.005`** | `0.173 ± 0.004` |
| Shared image latents ↑               | `0.67 ± 0.04`    | **`0.751 ± 0.07`** |
| Shared image latents on shape ↓      | `0.52 ± 0.005`   | **`0.289 ± 0.008`** |
| Shared image latents on noise ↓      | `0.035 ± 0.003`  | **`0.015 ± 0.007`** |

---

## Table 8: Behavioral decoding accuracy with various latent dimensions

| Latent Dimension | 50             | 80             | 100            |
|------------------|----------------|----------------|----------------|
| Shared neural latent | `0.41 ± 0.05` | `0.43 ± 0.03` | `0.42 ± 0.04` |

---

### Meta-Review · Area_Chair_ru3V · 2024-12-23

**Metareview:**

This paper introduces a novel "disentangling autoencoder" to identify shared and private latent features from multi-modal neural and behavioral data. The paper contributes to a growing literature on learning shared representations for multimodal data in neuroscience, where many researchers are interested in learning joint representations of brain data and behavior. The authors provided a great many clarifications and additional quantitive comparisons during the rebuttal period, and -- despite some disagreement amongst the four reviewers -- I am ultimately convinced that the novelty of the theoretical contribution (in particular, the orthogonality of its latent representations and the fact that it can find shared representations between 3 or more modalities), as well as the rigour of the comparisons to previous methods place it above the bar for acceptance.  Congratulations!  Please revise carefully to address all comments and add clarifications that arose during the discussion period.

**Additional Comments On Reviewer Discussion:**

This paper generated extensive discussion amongst reviewers, and the two reviewers who participated most actively in the discussions were ultimately split on whether it should be accepted or not. The reviewers raised some key issues of clarity and missing comparisons from the originally submitted version of the manuscript. However, the authors added a great deal of clarifying information that convinced me that the theoretical approach is sufficiently different from previous work to represent a worthwhile contribution to ICLR. And although reviewer QBbN raised concerns that the quantitative improvement over previous work was not impressive enough to represent a major advance, I was ultimately persuaded by the enthusiasm of reviewers HB99 and h2Dv, and by my own impressions of the additional comparisons provided during the rebuttal period.

---

### Decision · Program_Chairs · 2025-01-22

Accept (Poster)